# COOPERATIVE GRAPH NEURAL NETWORKS

## ABSTRACT

Graph neural networks are popular architectures for graph machine learning, based on iterative computation of node representations of an input graph through a series of invariant transformations. A large class of graph neural networks follow a standard message-passing paradigm: at every layer, each node state is updated based on an aggregate of messages from its neighborhood. In this work, we propose a novel framework for training graph neural networks, where every node is viewed as a *player* that can choose either 'listen', 'broadcast', 'listen and broadcast', or to 'isolate'. The standard message propagation scheme can then be viewed as a special case of this framework where every node 'listens and broadcasts' to all neighbors. Our approach offers a more flexible and dynamic message-passing paradigm, where each node can determine its own strategy based on their state, effectively exploring the graph topology while learning. We provide a theoretical analysis of the new message-passing scheme which is further supported by an extensive empirical analysis on a synthetic dataset and on real-world datasets.

## 1 INTRODUCTION

Graph neural networks (GNNs) (Scarselli et al., 2009; Gori et al., 2005) are a class of deep learning architectures for learning on graph-structured data. Their success in various graph machine learning tasks (Shlomi et al., 2021; Duvenaud et al., 2015; Zitnik et al., 2018) has led to a surge of different architectures (Kipf & Welling, 2017; Xu et al., 2019; Veličković et al., 2018; Hamilton et al., 2017; Li et al., 2016). GNNs are based on an iterative computation of node representations of an input graph through a series of invariant transformations. Gilmer et al. (2017) showed that the vast majority of GNNs can be implemented through *message-passing*, where the fundamental idea is to update each node's representation based on an aggregate of messages flowing from the node's neighbors.

The message-passing paradigm has been very influential in graph ML, but it also comes with well-known limitations related to the information flow on a graph, pertaining to *long-range* dependencies Dwivedi et al. (2022). In order to receive information from $k$-hop neighbors, a message-passing neural network needs at least $k$ layers. In many types of graphs, this typically implies an exponential growth of a node's receptive field. The growing amount of information needs then to be compressed into fixed-sized node embeddings, possibly leading to information loss, referred to as *over-squashing* (Alon & Yahav, 2021).

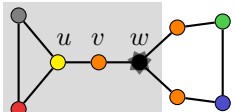

Figure 1: Node $w$ listens only to $v$. Applying the same choice iteratively, $w$ receives information from a subgraph marked in gray.

**Motivation.** Our goal is to generalize the message-passing scheme by allowing each node to decide how to propagate information *from* or *to* its neighbors, thus enabling a more flexible flow of information. As a motivating example, consider the graph depicted in Figure 1 and suppose that, at every layer, the black node $w$ needs information *only* from the neighbors which have a yellow neighbor (node $v$), and hence only information from a subgraph (highlighted with gray color). Such a scenario falls outside the ability of standard message-passing schemes because there is no mechanism to condition the propagation of information on the two-hop information.

The information flow becomes more complex if the nodes have different choices across layers, since we cannot anymore view the process as merely focusing on a subgraph. This kind of message-passing is then *dynamic* across layers. Consider the example depicted in Figure 2: the top row shows the information flow relative to the node $u$ across three layers, and the bottom row shows the information flow relative to the node $v$ across three layers. Node $u$ listens to every neighbor in the first layer, only

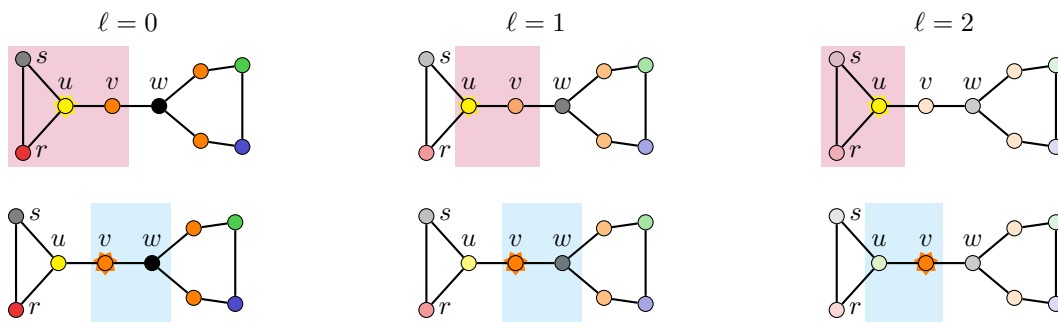

Figure 2: Example information flow for two nodes $u, v$. **Top**: information flow relative to $u$ across three layers. Node $u$ listens to every neighbor in the first layer, but only to $v$ in the second layer, and only to $s$ and $r$ in the last layer. **Bottom**: information flow relative to $v$ across three layers. The node $v$ listens only to $w$ in the first two layers, and only to $u$ in the last layer.

to $v$ in the second layer, and to nodes $s$ and $r$ in the last layer. On the other hand, node $v$ listens to node $w$ for the first two layers, and to node $u$ in the last layer.

**Main idea.** In the above examples, for each node, we need to learn whether or not to listen to a particular node in the neighborhood. To achieve this, we regard each node as a *player* that can take the following actions in each layer:

- STANDARD (S): Broadcast to neighbors that listen *and* listen to neighbors that broadcast.
- LISTEN (L): Listen to neighbors that broadcast.
- BROADCAST (B): Broadcast to neighbors that listen.
- ISOLATE (I): Neither listen nor broadcast, effectively isolating the node.

The case where all nodes perform the action STANDARD corresponds to the standard message-passing scheme used in GNNs. Conversely, having all the nodes ISOLATE corresponds to removing all the edges from the graph implying node-wise predictions. The interplay between these actions and the ability to change them *locally* and *dynamically* makes the overall approach richer and allows to decouple the input graph from the computational one and incorporate directionality into message-passing: a node can only listen to those neighbors that are currently broadcasting, and vice versa. We can emulate the example from Figure 2 by making $u$ choose the actions $\langle L, L, S \rangle$, $v$ and $w$ the actions $\langle S, S, L \rangle$, and $s$ and $r$ the actions $\langle S, I, S \rangle$.

**Contributions.** In this paper, we develop a new class of architectures, dubbed *cooperative graph neural networks* (CO-GNNs), where every node in the graph is viewed as a player that can perform one of the aforementioned actions. CO-GNNs comprise two jointly trained cooperating message-passing neural networks: an *environment network* $\eta$ (for solving the given task), and an *action network* $\pi$ (for choosing the best actions). Our contributions can be summarized as follows:

- We propose a novel, flexible message-passing mechanism for graph neural networks, which leads to CO-GNN architectures that effectively explore the graph topology while learning (Section 4).
- We provide a detailed discussion on the properties of CO-GNNs (Section 5.1) and show that they are more expressive than 1-dimensional Weisfeiler-Leman algorithm (Section 5.2), and more importantly, better suited for long-range tasks due to their adaptive nature (Section 5.3).
- Empirically, we focus on CO-GNNs with basic action and environment networks to carefully assess the virtue of the new message-passing paradigm. We first experimentally validate the strength of our approach on a synthetic task (Section 6.1). Afterwards, we conduct experiments on real-world datasets, and observe that CO-GNNs always improve compared to their baseline models, and further yield multiple state-of-the-art results (Sections 6.2 and 6.3). We complement these with further experiments reported in the Appendix. Importantly, we illustrate the dynamic and adaptive nature of CO-GNNs (Appendix B) and also provide experiments to evaluate CO-GNNs on long-range tasks (Appendix D).

Proofs of the technical results and additional experimental details can be found in the Appendix.

## 2 BACKGROUND

**Graph neural networks.** We consider simple, undirected attributed graphs $G = (V, E, \boldsymbol{X})$, where $\boldsymbol{X} \in \mathbb{R}^{|V| \times d}$ is a matrix of (input) node features, and $\boldsymbol{x}_v \in \mathbb{R}^d$ denotes the feature of a node $v \in V$. We focus on *message-passing neural networks (MPNNs)* (Gilmer et al., 2017) that encapsulate the vast majority of GNNs. An MPNN updates the initial node representations $\boldsymbol{h}_v^{(0)} = \boldsymbol{x}_v$ of each node $v$ for $0 \leq \ell \leq L - 1$ iterations based on its own state and the state of its neighbors $\mathcal{N}_v$ as:

$$\boldsymbol{h}_v^{(\ell+1)} = \phi^{(\ell)} \left( \boldsymbol{h}_v^{(\ell)}, \psi^{(\ell)} \left( \boldsymbol{h}_v^{(\ell)}, \{\!\!\{ \boldsymbol{h}_u^{(\ell)} \mid u \in \mathcal{N}_v \}\!\!\} \right) \right),$$

where $\{\!\!\{ \cdot \}\!\!\}$ denotes a multiset and $\phi^{(\ell)}$ and $\psi^{(\ell)}$ are differentiable *update* and *aggregation* functions, respectively. We denote by $d^{(\ell)}$ the dimension of the node embeddings at iteration (layer) $\ell$. The final representations $\boldsymbol{h}_v^{(L)}$ of each node $v$ can be used for predicting node-level properties or they can be pooled to form a graph embedding vector $\boldsymbol{z}_G^{(L)}$, which can be used for predicting graph-level properties. The pooling often takes the form of simple averaging, summation, or element-wise maximum. In this paper, we largely focus on the basic MPNNs of the following form:

$$\boldsymbol{h}_v^{(\ell+1)} = \sigma \left( \boldsymbol{W}_s^{(\ell)} \boldsymbol{h}_v^{(\ell)} + \boldsymbol{W}_n^{(\ell)} \psi \left( \{\!\!\{ \boldsymbol{h}_u^{(\ell)} \mid u \in \mathcal{N}_v \}\!\!\} \right) \right),$$

where $\boldsymbol{W}_s^{(\ell)}$ and $\boldsymbol{W}_n^{(\ell)}$ are $d^{(\ell)} \times d^{(\ell+1)}$ learnable parameter matrices acting on the node's self-representation and on the aggregated representation of its neighbors, respectively, $\sigma$ is a non-linearity, and $\psi$ is either *mean* or *sum* aggregation function. We refer to the architecture with mean aggregation as MEANGNNs and to the architecture with sum aggregation as SUMGNNs (Hamilton, 2020). We also consider prominent models such as GCN (Kipf & Welling, 2017) and GIN (Xu et al., 2019).

**Straight-through Gumbel-softmax estimator.** In our approach, we rely on an action network for predicting categorical actions for the nodes in the graph, which is not differentiable and poses a challenge for gradient-based optimization. One prominent approach to address this is given by the Gumbel-softmax estimator (Jang et al., 2017; Maddison et al., 2017) which effectively provides a differentiable, continuous approximation of discrete action sampling. Consider a finite set $\Omega$ of actions. We are interested in learning a categorical distribution over $\Omega$, which can be represented in terms of a probability vector $\boldsymbol{p} \in \mathbb{R}^{|\Omega|}$, whose elements store the probabilities of different actions. Let us denote by $\boldsymbol{p}(a)$ the probability of an action $a \in \Omega$. Gumbel-softmax is a special reparametrization trick that estimates the categorical distribution $\boldsymbol{p} \in \mathbb{R}^{|\Omega|}$ with the help of a Gumbel-distributed vector $\boldsymbol{g} \in \mathbb{R}^{|\Omega|}$, which stores an i.i.d. sample $\boldsymbol{g}(a) \sim \text{GUMBEL}(0, 1)$ for each action $a$. Given a categorical distribution $\boldsymbol{p}$ and a temperature parameter $\tau$, Gumbel-softmax scores can be computed as follows:

$$\text{Gumbel-softmax} \, (\boldsymbol{p}; \tau) = \frac{\exp\left((\log(\boldsymbol{p}) + \boldsymbol{g})/\tau\right)}{\sum_{a \in \Omega} \exp\left((\log(\boldsymbol{p}(a)) + \boldsymbol{g}(a))/\tau\right)}$$

As the softmax temperature $\tau$ decreases, the resulting vector tends to a *one-hot* vector. Straight-through Gumbel-softmax estimator utilizes the Gumbel-softmax estimator during the backward pass only (for a differentiable update), while during the forward pass, it employs an ordinary sampling.

## 3 RELATED WORK

Most of GNNs used in practice are instances of MPNNs (Gilmer et al., 2017) based on the message-passing approach, which has roots in classical GNN architectures (Scarselli et al., 2009; Gori et al., 2005) and their modern variations (Kipf & Welling, 2017; Xu et al., 2019; Veličković et al., 2018; Hamilton et al., 2017; Li et al., 2016).

Despite their success, MPNNs have some known limitations. First of all, their expressive power is upper bounded by the 1-dimensional Weisfeiler-Leman graph isomorphism test (1-WL) (Xu et al., 2019; Morris et al., 2019) in that MPNNs cannot distinguish any pair of graphs which cannot be distinguished by 1-WL. This drawback has motivated the study of more expressive architectures, based on higher-order graph neural networks (Morris et al., 2019; Maron et al., 2019; Keriven & Peyré, 2019), subgraph sampling approaches (Bevilacqua et al., 2022; Thiede et al., 2021), lifting graphs to higher-dimensional topological spaces (Bodnar et al., 2021), enriching the node features with unique

identifiers (Bouritsas et al., 2022; Loukas, 2020; You et al., 2021) or random features (Abboud et al., 2021; Sato et al., 2021). Second, MPNNs generally perform poorly on long-range tasks due to their information propagation bottlenecks (Li et al., 2018; Alon & Yahav, 2021). This motivated approaches based on rewiring the input graph (Klicpera et al., 2019; Topping et al., 2022; Karhadkar et al., 2023) by connecting relevant nodes and shortening propagation distances to minimize bottlenecks, or designing new message-passing architectures that act on distant nodes directly, e.g., using shortest-path distances (Abboud et al., 2022; Ying et al., 2021). Lately, there has been a surging interest in the advancement of Transformer-based approaches for graphs (Ma et al., 2023; Ying et al., 2021; Yun et al., 2019; Kreuzer et al., 2021; Dwivedi & Bresson, 2021), which can encompass complete node connectivity beyond the local information classical MPNNs capture, which in return, allows for more effective modeling of long-range interactions. Finally, classical message passing updates the nodes in a synchronous manner, which does not allow the nodes to react to messages from their neighbors individually. This has been recently argued as yet another limitation of classical message passing from the perspective of algorithmic alignment (Faber & Wattenhofer, 2022).

Through a new message-passing scheme, our work presents new perspectives on these limitations by dynamically changing the information flow depending on the task, and resulting in more flexible architectures than the classical MPNNs, while also allowing asynchronous message passing across nodes. Our work is related to the earlier work of Lai et al. (2020), where the goal is to update each node using a potentially different number of layers, which is achieved by learning the optimal aggregation depth for each node through a reinforcement learning approach. CO-GNNs are orthogonal to this study both in terms of the objectives and the approach (as detailed in Section 5).

## 4   COOPERATIVE GRAPH NEURAL NETWORKS

CO-GNNs view each node in a graph as *players* of a multiplayer environment, where the state of each player is given in terms of the representation (or *state*) of its corresponding node. In this environment, every node is updated following a two-stage process. In the first stage, each chooses an action from the set of actions given their current state and the states of their neighboring nodes. In the second stage, every node state gets updated based on their current state and the states of a *subset* of the neighboring nodes, as determined by the actions in the first stage. As a result, every node can determine how to propagate information from or to its neighbors.

Formally, a CO-GNN $(\pi, \eta)$ architecture is given in terms of two cooperating GNNs: (i) an action network $\pi$ for choosing the best actions, and (ii) an environment network $\eta$ for updating the node representations. A CO-GNN layer updates the representations $\boldsymbol{h}_v^{(\ell)}$ of each node $v$ as follows. First, an action network $\pi$ predicts, for each node $v$, a probability distribution $\boldsymbol{p}_v^{(\ell)} \in \mathbb{R}^4$ over the actions $\{\mathrm{S}, \mathrm{L}, \mathrm{B}, \mathrm{I}\}$ that $v$ can take, given its state and the state of its neighbors $\mathcal{N}_v$, as follows:

$$\boldsymbol{p}_v^{(\ell)} = \pi\left(\boldsymbol{h}_v^{(\ell)}, \{\!\!\{\boldsymbol{h}_u^{(\ell)} \mid u \in \mathcal{N}_v\}\!\!\}\right). \tag{1}$$

Then, for each node $v$, an action is sampled $a_v^{(\ell)} \sim \boldsymbol{p}_v^{(\ell)}$ using the Straight-through Gumbel-softmax estimator, and an environment network $\eta$ is utilized to update the state of each node in accordance with the sampled actions, as follows:

$$\boldsymbol{h}_v^{(\ell+1)} = \begin{cases} \eta^{(\ell)}\left(\boldsymbol{h}_v^{(\ell)}, \{\!\!\{\}\!\!\}\right), & a_v^{(\ell)} = \mathrm{I} \vee \mathrm{B} \\ \eta^{(\ell)}\left(\boldsymbol{h}_v^{(\ell)}, \{\!\!\{\boldsymbol{h}_u^{(\ell)} \mid u \in \mathcal{N}_v, a_u^{(\ell)} = \mathrm{S} \vee \mathrm{B}\}\!\!\}\right), & a_v^{(\ell)} = \mathrm{L} \vee \mathrm{S}. \end{cases} \tag{2}$$

This corresponds to a single layer update, and, as usual, by stacking $L \geq 1$ layers, we obtain the representations $\boldsymbol{h}_v^{(L)}$ for each node $v$. In its full generality, a CO-GNN $(\pi, \eta)$ architecture can use any GNN architecture in place of the action network $\pi$ and the environment network $\eta$. For the sake of our study, we focus on simple models such as SUMGNNs, MEANGNNs, GCN, and GIN, which are respectively denoted as $\sum, \mu, *$ and $\epsilon$. For example, we write CO-GNN$(\Sigma, \mu)$ to denote a CO-GNN architecture which uses SUMGNN as its action network and MEANGNN as its environment network.

Fundamentally, CO-GNNs update the node states in a fine-grained manner as formalized in Equation (2): if a node $v$ chooses to ISOLATE or to BROADCAST then it gets updated only based on its previous state, which corresponds to a node-wise update function. On the other hand, if a node $v$ chooses the action LISTEN or STANDARD then it gets updated based on its previous state as well as the state of its neighbors which perform the actions BROADCAST or STANDARD at this layer.

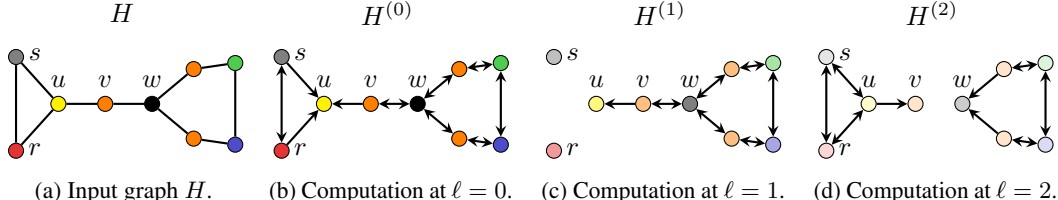

| $H$ | $H^{(0)}$ | $H^{(1)}$ | $H^{(2)}$ |
|---|---|---|---|
| (a) Input graph $H$. | (b) Computation at $\ell = 0$. | (c) Computation at $\ell = 1$. | (d) Computation at $\ell = 2$. |

Figure 3: The input graph $H$ and its computation graphs $H^{(0)}$, $H^{(1)}$, $H^{(2)}$ at the respective layers. The computation graphs are a result of applying the following actions: $\langle L, L, S \rangle$ for the node $u$; $\langle S, S, L \rangle$ for the nodes $v$ and $w$; $\langle S, I, S \rangle$ for the nodes $s$ and $r$; $\langle S, S, S \rangle$ for all other nodes.

## 5 MODEL PROPERTIES

In this section, we provide a detailed analysis of CO-GNNs, focusing on its conceptual novelty, its expressive power, and its suitability to long-range tasks.

### 5.1 CONCEPTUAL PROPERTIES OF THE LEARNABLE MESSAGE-PASSING PARADIGM

**Task-specific**: Standard message-passing updates nodes based on their local neighborhood, which is completely task-agnostic. By allowing each node to listen to the information only from 'relevant' neighbors, CO-GNNs can determine a computation graph which is best suited for the target task. For example, if the task requires information only from the neighbors with a certain degree then the action network can learn to listen only to these nodes, as we experimentally validate in Section 6.1.

**Directed**: The outcome of the actions that the nodes can take amounts to a special form of *'directed rewiring'* of the input graph: an edge can be *dropped* (e.g., if two neighbors listen without broadcasting); an edge can remain *undirected* (e.g., if both neighbors apply the standard action); or, an edge can *become directed* implying directional information flow (e.g., if one neighbor listens while its neighbor broadcasts). Taking this perspective, the proposed message-passing can be seen as operating on a potentially different directed graph induced by the choice of actions at every layer. Formally, given a graph $G = (V, E)$, let us denote by $G^{(\ell)} = (V, E^{(\ell)})$ the directed computational graphs induced by the actions chosen at layer $\ell$, where $E^{(\ell)}$ is the set of directed edges at layer $\ell$. We can rewrite the update given in Equation (2) concisely as follows:

$$\boldsymbol{h}_v^{(\ell+1)} = \eta^{(\ell)} \left( \boldsymbol{h}_v^{(\ell)}, \{\!\!\{ \boldsymbol{h}_u^{(\ell)} \mid (u, v) \in E^{(\ell)} \}\!\!\} \right).$$

Consider the input graph $H$ from Figure 3: $u$ gets messages from $v$ only in the first two layers, and $v$ gets messages from $u$ only in the last layer, illustrating a directional message-passing between these nodes. This abstraction allows for a direct implementation of CO-GNNs by simply considering the induced graph adjacency matrix at every layer.

**Dynamic**: In this setup, each node learns to interact with the 'relevant' neighbors and does so only as long as they remain relevant: CO-GNNs do not operate on a pre-fixed computational graph, but rather on a learned computational graph, which is dynamic across layers. In our running example, observe that the computational graph is a different one at every layer (depicted in Figure 4). This brings advantages for the information flow as we outline in Section 5.3.

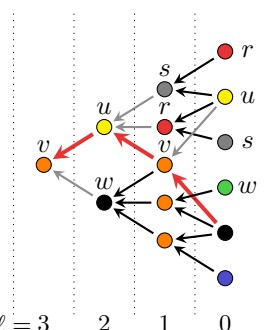

**Feature and structure based**: Standard message-passing is completely determined by the structure of the graph: two nodes with the same neighborhood get the same aggregated message. This is not necessarily the case in our setup, since the action network can learn different actions for two nodes with different node features, e.g., by choosing different actions for a *red* node and a *blue* node. This enables different messages for different nodes even if their neighborhoods are identical.

Figure 4: The computational graph of $H$.

**Asynchronous**: Standard message-passing updates all nodes synchronously at every iteration, which is not always optimal as argued by Faber & Wattenhofer (2022), especially when the task requires to treat the nodes non-uniformly. By design, CO-GNNs enable asynchronous updates across nodes.

**Efficient**: While being more sophisticated, the proposed message-passing algorithm is efficient in terms of runtime, as we detail in Appendix C. CO-GNNs are also parameter-efficient: they use the same policy network and as a result a comparable number of parameters to their baseline models.

## 5.2 EXPRESSIVE POWER OF COOPERATIVE GRAPH NEURAL NETWORKS

The environment and action networks of CO-GNN architectures are parameterized by standard MPNNs. This raises an obvious question regarding the expressive power of CO-GNN architectures: *are* CO-GNN*s also bounded by 1-WL?* Consider, for instance, the non-isomorphic graphs $G_1$ and $G_2$ depicted in Figure 5. Standard MPNNs cannot distinguish these graphs, while CO-GNNs can:

**Proposition 5.1.** *Let $G_1 = (V_1, E_1, \boldsymbol{X}_1)$ and $G_2 = (V_2, E_2, \boldsymbol{X}_2)$ be two non-isomorphic graphs. Then, for any threshold $0 < \delta < 1$, there exists a parametrization of a CO-GNN architecture using sufficiently many layers $L$, satisfying $\mathbb{P}(\boldsymbol{z}_{G_1}^{(L)} \neq \boldsymbol{z}_{G_2}^{(L)}) \geq 1 - \delta$.*

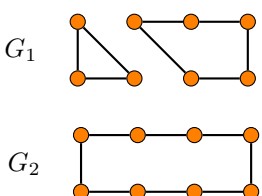

The explanation for this result is the following: CO-GNN architectures learn, at every layer, and for each node $u$, a probability distribution over the actions. These learned distributions are identical for two isomorphic nodes. However, the process relies on *sampling* actions from these distributions, and clearly, the samples from identical distributions can differ. This makes CO-GNN models *invariant in expectation,* and the variance introduced by the sampling process helps to discriminate nodes that are 1-WL indistinguishable. Thus, for two nodes indistinguishable by 1-WL, there is a non-trivial probability of sampling a different action for the respective nodes, which in turn makes their direct neighborhood differ. This yields unique node identifiers (see, e.g., (Loukas, 2020)) with high probability and allows us to distinguish any pair of graphs assuming an injective graph pooling function (Xu

Figure 5: $G_1$ and $G_2$ are indistinguishable by a wide class of GNNs. CO-GNNs can distinguish these graphs.

et al., 2019). Our result is analogous to GNNs with random node features (Abboud et al., 2021; Sato et al., 2021), which are more expressive than their classical counterparts. Another relation is to subgraph GNNs (Bevilacqua et al., 2022; Papp et al., 2021), which bypass 1-WL expressiveness limitation by considering subgraphs extracted from $G$. In our framework, the sampling of different actions has an analogous effect. Note that CO-GNNs are *not* designed for more expressive power, and our result relies merely on variations in the sampling process, which should be noted as a limitation. We validate the stated expressiveness gain on a synthetic experiment in Appendix D.

## 5.3 DYNAMIC MESSAGE-PASSING FOR LONG-RANGE TASKS

Long-range tasks necessitate to propagate information between distant nodes: we argue that a dynamic message-passing paradigm is highly effective for such tasks since it becomes possible to propagate only relevant task-specific information. Suppose, for instance, that we are interested in transmitting information from a source node to a distant target node: our message-passing paradigm can efficiently filter irrelevant information by learning to focus on the shortest path connecting these two nodes, hence maximizing the information flow to the target node. We can generalize this observation towards receiving information from multiple distant nodes and prove the following:

**Theorem 5.2.** *Let $G = (V, E, \boldsymbol{X})$ be a connected graph with node features. For some $k > 0$, for any target node $v \in V$, for any $k$ source nodes $u_1, \dots, u_k \in V$, and for any compact, differentiable function $f : \mathbb{R}^{d^{(0)}} \times \dots \times \mathbb{R}^{d^{(0)}} \to \mathbb{R}^d$, there exists an $L$-layer CO-GNN computing final node representations such that for any $\epsilon, \delta > 0$ it holds that $\mathbb{P}(|\boldsymbol{h}_v^{(L)} - f(\boldsymbol{x}_{u_1}, \dots \boldsymbol{x}_{u_k})| < \epsilon) \geq 1 - \delta$.*

This means that if a property of a node $v$ is a function of $k$ distant nodes then CO-GNNs can approximate this function. This follows from two findings: (i) the features of $k$ nodes can be transmitted to the source node without loss of information in CO-GNNs and (ii) the final layer of a CO-GNN architecture, e.g., an MLP, can approximate any differentiable function over $k$ node features (Hornik, 1991; Cybenko, 1989). We validate these findings empirically on long-range interactions datasets (Dwivedi et al., 2022) in Appendix D.

## 5.4 OVER-SQUASHING AND OVER-SMOOTHING

**Over-squashing** refers to the failure of message passing to propagate information on the graph. Topping et al. (2022); Di Giovanni et al. (2023) formalized over-squashing as the insensitivity of an $r$-layer MPNN output at node $u$ to the input features of a distant node $v$, expressed through a bound on the Jacobian $\|\partial \boldsymbol{h}_v^{(r)}/\partial \boldsymbol{x}_u\| \leq C^r(\hat{\boldsymbol{A}}^r)_{vu}$, where $C$ encapsulated architecture-related constants (e.g., width, smoothness of the activation function, etc.) and the normalized adjacency matrix $\hat{\boldsymbol{A}}$ captures the effect of the graph. Graph rewiring techniques amount to modifying $\hat{\boldsymbol{A}}$ so as to increase the upper bound and thereby reduce the effect of over-squashing.

Since the actions of every node in CO-GNNs result in an effective graph rewiring (different at every layer), the action network can choose actions that transmit the features of node $u \in V$ to node $v \in V$ as shown in Theorem 5.2, resulting in the maximization of the bound on the Jacobian.

**Over-smoothing** refers to the tendency of node embeddings to become increasingly similar across the graph with the increase in the number of message passing layers (Li et al., 2018). Recently, Rusch et al. (2022) showed that over-smoothing can be mitigated through the gradient gating mechanism, which adaptively disables the update of a node from neighbors with similar features. Our architecture, through the choice of BROADCAST or ISOLATE actions, allows to mimic this mechanism.

## 6 EXPERIMENTAL RESULTS

We evaluate CO-GNNs on (i) a synthetic experiment comparing with classical MPNNs, (ii) real-world node classification datasets (Platonov et al., 2023), and (iii) real-world graph classification datasets (Morris et al., 2020). We provide a detailed analysis regarding the actions learned by CO-GNNs on different datasets in Appendix B, illustrating the task-specific nature of CO-GNNs. Finally, we report a synthetic expressiveness experiment and an experiment on long-range interactions datasets (Dwivedi et al., 2022) in Appendix D.

**Implementation.** We trained and tested our model on NVidia GTX V100 GPUs with Python 3.9, Cuda 11.8, PyTorch 2.0.0, and PyTorch Geometric (Fey & Lenssen, 2019).

### 6.1 SYNTHETIC EXPERIMENT ON ROOTNEIGHBORS

**Task.** In this experiment, we compare CO-GNNs to a class of MPNNs on a new dataset: ROOTNEIGHBORS. Specifically, we consider the following regression task: *given a rooted tree, predict the average of the features of root-neighbors of degree* 6. This is an intricate task since it requires first identifying the neighbors of the root node with degree 6 and then returning the average feature of these nodes as the target value. ROOTNEIGHBORS consists of trees of depth 2 with random node features of dimension $d = 5$. The data generation ensures, for each root in each tree has at least one neighbor which is of degree 6 (detailed in Appendix E.2). One example tree is shown in Figure 6a: the root node $r$ has only two neighbors with degree 6 ($u$ and $v$) and the target prediction value is $(\boldsymbol{x}_u + \boldsymbol{x}_v)/2$.

**Setup.** We trained GCN (Kipf & Welling, 2017), Graph-SAGE (Hamilton et al., 2017), GAT (Veličković et al., 2018), SUMGNN, MEANGNN, as baselines and compared them to CO-GNN($\Sigma, \Sigma$), CO-GNN($\mu, \mu$) and CO-GNN($\Sigma, \mu$). We used the Adam optimizer and reported all hyperparameters in the appendix (Table 8). We report the Mean Average Error (MAE).

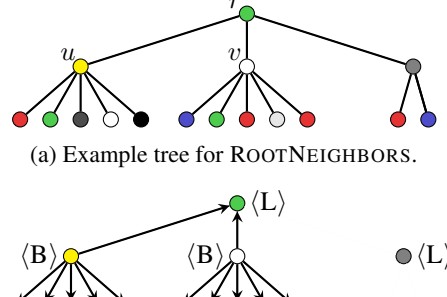

(a) Example tree for ROOTNEIGHBORS.

(b) Example of an optimal directed subgraph, where the nodes with a degree of 6 ($v$ and $w$) BROADCAST, while other nodes LISTEN.

Figure 6: ROOTNEIGHBORS example.

**Results for MPNNs.** The results are presented in Table 1, which includes the random baseline: the average MAE obtained via a random prediction. All MPNNs perform poorly on this task. GCN, GAT, and MEANGNN fail to identify node degrees, making it impossible to detect nodes with a

specific degree, which is crucial for the task. GCN and GAT are only marginally better than the random baseline, whereas MEANGNN performs substantially better than the random baseline. The latter can be explained by the fact that MEANGNN employs a different transformation on the source node rather than treating it as a neighbor (unlike the self-loop in GCN/GAT) and this yields better average MAE. SAGE and SUMGNN use sum aggregation and can identify the node degrees, but they struggle in averaging the node features, which yield comparable MAE results to that of MEANGNN.

**Results for CO-GNNs.** The ideal mode of operation for CO-GNNs would be as follows:

1. The action network chooses either the action LISTEN or STANDARD for the root node, and the action BROADCAST or STANDARD for the root-neighbors which have a degree 6,

2. The action network chooses either the action LISTEN or the action ISOLATE for all the remaining root-neighbors, and

3. The environment network updates the root node by averaging the features of its neighbors which are currently broadcasting.

This is depicted in Figure 6 and can be achieved with a single-layer CO-GNN assuming (1)-(3) can be accomplished. The best result is achieved by CO-GNN$(\Sigma, \mu)$, because SUMGNN (as the action network) can accomplish (1) and (2), and MEANGNN (as the environment network) can accomplish (3). This model leverages the strengths of the SUMGNN model and the MEANGNN model to cater to the different roles of the action and environment networks, making it the most natural CO-GNN model for the regression task. The next best model is CO-GNN$(\Sigma, \Sigma)$, which also uses SUMGNN as the action network, accomplishing (1) and (2). However, it uses another SUMGNN as the environment network which cannot easily mimic the averaging of the neighbor's features. Finally, CO-GNN$(\mu, \mu)$ performs weakly, since MEANGNN as an action network cannot achieve (1) hindering the performance of the whole task. Indeed, CO-GNN$(\mu, \mu)$ performs comparably to MEANGNN suggesting that the action network is not useful in this case.

Table 1: Results on ROOT-NEIGHBORS. Top three models are colored by First, Second, Third.

| Model | MAE |
|---|---|
| Random | 0.474 |
| GCN | 0.468 |
| SAGE | 0.336 |
| GAT | 0.442 |
| SUMGNN | 0.370 |
| MEANGNN | 0.329 |
| CO-GNN$(\Sigma, \Sigma)$ | 0.196 |
| CO-GNN$(\mu, \mu)$ | 0.339 |
| CO-GNN$(\Sigma, \mu)$ | 0.079 |

To shed light on the performance of CO-GNN models, we computed the percentage of edges which are accurately retained or removed by the action network in a single layer CO-GNN model. We observe an accuracy of 57.20% for CO-GNN$(\mu, \mu)$, 99.55% for CO-GNN$(\Sigma, \Sigma)$, and 99.71% for CO-GNN$(\Sigma, \mu)$, which empirically confirms the expected behavior of CO-GNNs. In fact, the example tree shown in Figure 6 is taken from the ROOTNEIGHBORS, and reassuringly, CO-GNN$(\Sigma, \mu)$ learns precisely the actions that induce the shown optimal subgraph.

## 6.2 NODE CLASSIFICATION

One of the strengths of CO-GNNs is their capability to utilize task-specific information propagation, which raises an obvious question: could CO-GNNs outperform the baselines on heterophilious graphs, where standard message passing is known to suffer? To answer this question, we assess the performance of CO-GNNs on heterophilic node classification datasets from (Platonov et al., 2023).

**Setup.** We evaluate CO-GNN$(\Sigma, \Sigma)$ and CO-GNN$(\mu, \mu)$ on the 5 heterophilic graphs, following the 10 data splits and the methodology of Platonov et al. (2023) and report the accuracy and standard deviation for roman-empire and amazon-ratings, and mean ROC AUC and standard deviation for minesweeper, tolokers, and questions. The classical baselines GCN (Kipf & Welling, 2017), GraphSAGE (Hamilton et al., 2017), GAT (Veličković et al., 2018), GAT-sep, GT (Shi et al., 2021) and GT-sep are from (Platonov et al., 2023). We use the Adam optimizer and report all hyperparameters in the Appendix E.4.

**Results.** All results are reported in Table 2: CO-GNNs achieve state-of-the-art across results the board, despite using relatively simple architectures as their action and environment networks. Overall, CO-GNNs demonstrate an average accuracy improvement of 2.23% compared to all baseline methods, across all datasets, surpassing the performance of more complex models such as GT. These results are reassuring as they establish CO-GNNs as a strong method in the heterophilic setting.

Table 2: Results on node classification. Top three models are colored by First, Second, Third.

| | roman-empire | amazon-ratings | minesweeper | tolokers | questions |
|---|---|---|---|---|---|
| GCN | $73.69 \pm 0.74$ | $48.70 \pm 0.63$ | $89.75 \pm 0.52$ | $83.64 \pm 0.67$ | $76.09 \pm 1.27$ |
| SAGE | $85.74 \pm 0.67$ | $53.63 \pm 0.39$ | $93.51 \pm 0.57$ | $82.43 \pm 0.44$ | $76.44 \pm 0.62$ |
| GAT | $80.87 \pm 0.30$ | $49.09 \pm 0.63$ | $92.01 \pm 0.68$ | $83.70 \pm 0.47$ | $77.43 \pm 1.20$ |
| GAT-sep | $88.75 \pm 0.41$ | $52.70 \pm 0.62$ | $93.91 \pm 0.35$ | $83.78 \pm 0.43$ | $76.79 \pm 0.71$ |
| GT | $86.51 \pm 0.73$ | $51.17 \pm 0.66$ | $91.85 \pm 0.76$ | $83.23 \pm 0.64$ | $77.95 \pm 0.68$ |
| GT-sep | $87.32 \pm 0.39$ | $52.18 \pm 0.80$ | $92.29 \pm 0.47$ | $82.52 \pm 0.92$ | $78.05 \pm 0.93$ |
| CO-GNN$(\Sigma, \Sigma)$ | $91.57 \pm 0.32$ | $51.28 \pm 0.56$ | $95.09 \pm 1.18$ | $83.36 \pm 0.89$ | $80.02 \pm 0.86$ |
| CO-GNN$(\mu, \mu)$ | $91.37 \pm 0.35$ | $54.17 \pm 0.37$ | $97.31 \pm 0.41$ | $84.45 \pm 1.17$ | $76.54 \pm 0.95$ |

Table 3: Results on graph classification. Top three models are colored by First, Second, Third.

| | IMDB-B | IMDB-M | REDDIT-B | NCI1 | PROTEINS | ENZYMES |
|---|---|---|---|---|---|---|
| DGCNN | $69.2 \pm 3.0$ | $45.6 \pm 3.4$ | $87.8 \pm 2.5$ | $76.4 \pm 1.7$ | $72.9 \pm 3.5$ | $38.9 \pm 5.7$ |
| DiffPool | $68.4 \pm 3.3$ | $45.6 \pm 3.4$ | $89.1 \pm 1.6$ | $76.9 \pm 1.9$ | $73.7 \pm 3.5$ | $59.5 \pm 5.6$ |
| ECC | $67.7 \pm 2.8$ | $43.5 \pm 3.1$ | OOR | $76.2 \pm 1.4$ | $72.3 \pm 3.4$ | $29.5 \pm 8.2$ |
| GIN | $71.2 \pm 3.9$ | $48.5 \pm 3.3$ | $89.9 \pm 1.9$ | $80.0 \pm 1.4$ | $73.3 \pm 4.0$ | $59.6 \pm 4.5$ |
| GraphSAGE | $68.8 \pm 4.5$ | $47.6 \pm 3.5$ | $84.3 \pm 1.9$ | $76.0 \pm 1.8$ | $73.0 \pm 4.5$ | $58.2 \pm 6.0$ |
| ICGMM$_f$ | $71.8 \pm 4.4$ | $49.0 \pm 3.8$ | $91.6 \pm 2.1$ | $76.4 \pm 1.4$ | $73.2 \pm 3.9$ | - |
| SPN$(k=5)$ | - | - | - | $78.6 \pm 1.7$ | $74.2 \pm 2.7$ | $69.4 \pm 6.2$ |
| CO-GNN$(\Sigma, \Sigma)$ | $70.8 \pm 3.3$ | $48.5 \pm 4.0$ | $88.6 \pm 2.2$ | $80.6 \pm 1.1$ | $73.1 \pm 2.3$ | $65.7 \pm 4.9$ |
| CO-GNN$(\mu, \mu)$ | $72.2 \pm 4.1$ | $49.9 \pm 4.5$ | $90.4 \pm 1.9$ | $79.4 \pm 0.7$ | $71.3 \pm 2.0$ | $68.3 \pm 5.7$ |

## 6.3 GRAPH CLASSIFICATION

In this experiment, we evaluate CO-GNNs on the TUDataset (Morris et al., 2020) graph classification benchmark.

**Setup.** We evaluate CO-GNN$(\Sigma, \Sigma)$ and CO-GNN$(\mu, \mu)$ on the 7 graph classification benchmarks, following the risk assessment protocol of Errica et al. (2020), and report the mean accuracy and standard deviation. The results for the baselines DGCNN (Wang et al., 2019), DiffPool (Ying et al., 2018), Edge-Conditioned Convolution (ECC) (Simonovsky & Komodakis, 2017), GIN (Xu et al., 2019), GraphSAGE (Hamilton et al., 2017) are from Errica et al. (2020). We also include ICGMM$_f$ (Castellana et al., 2022), and SPN$(k = 5)$ (Abboud et al., 2022) as more recent baselines. OOR (Out of Resources) implies extremely long training time or GPU memory usage. We use Adam optimizer and StepLR learn rate scheduler, and report all hyperparameters in the appendix (Table 11).

**Results.** CO-GNN models achieve the highest accuracy on three datasets out of six as reported in Table 3 and remain competitive on the other datasets. CO-GNN yield these performance improvements, despite using relatively simple action and environment networks, which is intriguing as CO-GNNs unlock a large design space which includes a large class of model variations.

## 7 CONCLUSIONS

We proposed a novel framework for training graph neural networks which entails a new dynamic message-passing scheme. We introduced the CO-GNN architectures, which can explore the graph topology while learning: this is achieved by a joint training regime including an *action network* $\pi$ and an *environment network* $\eta$. Our work brings in novel perspectives particularly pertaining to the well-known limitations of GNNs. We have empirically validated our theoretical findings on real-world datasets and synthetic datasets and provided insights regarding the model behavior. We think that this work brings novel perspectives to graph machine learning which need to be further explored in future work.

## 8 REPRODUCIBILITY STATEMENT

To ensure the reproducibility of this paper, we include the Appendix with five main sections. Appendix A includes detailed proofs for the technical statements presented in the paper. Appendix E provides data generation protocol for ROOTNEIGHBORS and further details of the real-world datasets that are used in Section 6. The experimental results in the paper are reproducible with the hyperparameter settings for all results contained in Tables 8 to 11 in Appendix E.4. The code to reproduce all experiments, along with the code to generate the datasets and tasks we propose is released at https://anonymous.4open.science/r/CoGNN.

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

# A  PROOFS OF TECHNICAL RESULTS

## A.1  PROOF OF PROPOSITION 5.1

In order to prove Proposition 5.1, we first prove the following lemma, which shows that all non-isolated nodes of an input graph can be individualized by CO-GNNs:

**Lemma A.1.** *Let $G = (V, E, \boldsymbol{X})$ be a graph with node features. For every pair of non-isolated nodes $u, v \in V$ and for all $\delta > 0$, there exists a CO-GNN architecture with sufficiently many layers $L$ which satisfies $\mathbb{P}(\boldsymbol{h}_u^{(L)} \neq \boldsymbol{h}_v^{(L)}) \geq 1 - \delta$.*

*Proof.* We consider an $L$-layer CO-GNN$(\eta, \pi)$ architecture satisfying the following:

    (i)  the *environment* network $\eta$ is composed of $L$ injective layers,

    (ii)  the *action* network $\pi$ is composed of a *single* layer, and it is shared across CO-GNN layers.

Item (i) can be satisfied by a large class of GNN architectures, including SUMGNN (Morris et al., 2019) and GIN (Xu et al., 2019). We start by assuming $\boldsymbol{h}_u^{(0)} = \boldsymbol{h}_v^{(0)}$. These representations can be differentiated if the model can jointly realize the following actions at some layer $\ell$ using the action network $\pi$:

    1.  $a_u^{(\ell)} = \text{L} \vee \text{S}$,

    2.  $a_v^{(\ell)} = \text{I} \vee \text{B}$, and

    3.  $\exists$ a neighbor $w$ of $u$ s.t. $a_w^{(\ell)} = \text{S} \vee \text{B}$.

The key point is to ensure an update for the state of $u$ via an aggregated message from its neighbors (at least one), while isolating $v$. In what follows, we assume the worst-case for the degree of $u$ and consider a node $w$ to be the only neighbor of $u$. Let us denote the joint probability of realizing these actions 1-3 for the nodes $u, v, w$ at layer $\ell$ as:

$$p_{u,v}^{(\ell)} = \mathbb{P}\left( (a_u^{(\ell)} = \text{L} \vee \text{S}) \wedge (a_v^{(\ell)} = \text{I} \vee \text{B}) \wedge (a_w^{(\ell)} = \text{S} \vee \text{B}) \right).$$

The probability of taking each action is non-zero (since it is a result of applying softmax) and $u$ has at least one neighbor (non-isolated), therefore $p_{u,v}^{(\ell)} > 0$. For example, if we assume a constant action network that outputs a uniform distribution over the possible actions (each action probability 0.25) then $p_{u,v}^{(\ell)} = 0.125$.

This means that the environment network $\eta$ applies the following updates to the states of $u$ and $v$ with probability $p_{u,v}^{(\ell)} > 0$:

$$\boldsymbol{h}_u^{(\ell+1)} = \eta^{(\ell)}\left( \boldsymbol{h}_u^{(\ell)}, \{\!\!\{\boldsymbol{h}_w^{(\ell)} \mid w \in \mathcal{N}_u, a_w^{(\ell)} = \text{S} \vee \text{B}\}\!\!\} \right),$$
$$\boldsymbol{h}_v^{(\ell+1)} = \eta^{(\ell)}\left( \boldsymbol{h}_v^{(\ell)}, \{\!\!\{\}\!\!\} \right).$$

The inputs to the environment network layer $\eta^{(\ell)}$ for these updates are clearly different, and since the environment layer is injective, we conclude that $\boldsymbol{h}_u^{(\ell+1)} \neq \boldsymbol{h}_v^{(\ell+1)}$.

Thus, the probability of having different final representations for the nodes $u$ and $v$ is lower bounded by the probability of the events 1-3 jointly occurring at least once in one of the CO-GNN layers, which, by applying the union bound, yields:

$$\mathbb{P}(\boldsymbol{h}_u^{(L_{u,v})} \neq \boldsymbol{h}_v^{(L_{u,v})}) \geq 1 - \prod_{\ell=0}^{L_{u,v}} \left(1 - p_{u,v}^{(\ell)}\right) \geq 1 - (1 - \gamma_{u,v})^{L_{u,v}} \geq 1 - \delta$$

where $\gamma_{u,v} = \max_{\ell \in [L_{u,v}]} \left( p_{u,v}^{(\ell)} \right)$ and $L_{u,v} = \log_{1-\gamma_{u,v}} (\delta)$.

We repeat this process for all pairs of non-isolated nodes $u, v \in V$. Due to the injectivity of $\eta^{(\ell)}$ for all $\ell \in [L]$, once the nodes are distinguished, they cannot remain so in deeper layers of the architecture, which ensures that all nodes $u, v \in V$ differ in their final representations $\boldsymbol{h}_u^{(\ell)} \neq \boldsymbol{h}_v^{(\ell)}$ after this process completes. The number of layers required for this construction is then given by:

$$L = |V \setminus I| \log_{1-\alpha}(\delta) \geq \sum_{u,v \in V \setminus I} \log_{1-\gamma_{u,v}}(\delta),$$

where $I$ is the set of all isolated nodes in $V$ and

$$\alpha = \max_{u,v \in V \setminus I}(\gamma_{u,v}) = \max_{u,v \in V \setminus I}\left(\max_{\ell \in [L_{u,v}]}\left(p_{u,v}^{(\ell)}\right)\right).$$

Having shown a CO-GNN construction with the number of layers bounded as above, we conclude the proof. $\qquad\square$

**Proposition 5.1.** *Let $G_1 = (V_1, E_1, \boldsymbol{X}_1)$ and $G_2 = (V_2, E_2, \boldsymbol{X}_2)$ be two non-isomorphic graphs. Then, for any threshold $0 < \delta < 1$, there exists a parametrization of a CO-GNN architecture using sufficiently many layers L, satisfying $\mathbb{P}(\boldsymbol{z}_{G_1}^{(L)} \neq \boldsymbol{z}_{G_2}^{(L)}) \geq 1 - \delta$.*

*Proof.* Let $\delta > 0$ be any value and consider the graph $G = (V, E, \boldsymbol{X})$ which has $G_1$ and $G_2$ as its components:

$$V = V_1 \cup V_2, \quad E = E_1 \cup E_2, \quad \boldsymbol{X} = \boldsymbol{X}_1 \| \boldsymbol{X}_2,$$

where $\|$ is the matrix horizontal concatenation. By Lemma A.1, for every pair of non-isolated nodes $u, v \in V$ and for all $\delta > 0$, there exists a CO-GNN architecture with sufficiently many layers $L = |V \setminus I| \log_{1-\alpha}(\delta)$ which satisfies:

$$\mathbb{P}(\boldsymbol{h}_u^{(L_{u,v})} \neq \boldsymbol{h}_v^{(L_{u,v})}) \geq 1 - \delta, \text{ with } \alpha = \max_{u,v \in V \setminus I}\left(\max_{\ell \in [L_{u,v}]}\left(p_{u,v}^{(\ell)}\right)\right),$$

where $p_{u,v}^{(\ell)}$ represents a lower bound on the probability for the representations of nodes $u, v \in V$ at layer $\ell$ being different.

We use the same CO-GNN construction given in Lemma A.1 on $G$, which ensures that all non-isolated nodes have different representations in $G$. When applying this CO-GNN to $G_1$ and $G_2$ separately, we get that every non-isolated node from either graph has a different representation with probability $1 - \delta$ as a result. Hence, the multiset $\mathcal{M}_1$ of node features for $G_1$ and the multiset $\mathcal{M}_2$ of node features of $G_2$ must differ. Assuming an injective pooling function from these multisets to graph-level representations, we get:

$$\mathbb{P}(\boldsymbol{z}_{G_1}^{(L)} \neq \boldsymbol{z}_{G_2}^{(L)}) \geq 1 - \delta$$

for $L = |V \setminus I| \log_{1-\alpha}(\delta)$. $\qquad\square$

## A.2 PROOF OF THEOREM 5.2

**Theorem 5.2.** *Let $G = (V, E, \boldsymbol{X})$ be a connected graph with node features. For some $k > 0$, for any target node $v \in V$, for any k source nodes $u_1, \ldots, u_k \in V$, and for any compact, differentiable function $f : \mathbb{R}^{d^{(0)}} \times \ldots \times \mathbb{R}^{d^{(0)}} \to \mathbb{R}^d$, there exists an L-layer CO-GNN computing final node representations such that for any $\epsilon, \delta > 0$ it holds that $\mathbb{P}(|\boldsymbol{h}_v^{(L)} - f(\boldsymbol{x}_{u_1}, \ldots \boldsymbol{x}_{u_k})| < \epsilon) \geq 1 - \delta$.*

*Proof.* For arbitrary $\epsilon, \delta > 0$, we start by constructing a feature encoder $\text{ENC} : \mathbb{R}^{d^{(0)}} \to \mathbb{R}^{2(k+1)d^{(0)}}$ which encodes the initial representations $\boldsymbol{x}_w \in \mathbb{R}^{d^{(0)}}$ of each node $w$ as follows:

$$\text{ENC}(\boldsymbol{x}_w) = \big[\underbrace{\tilde{\boldsymbol{x}}_w^\top \oplus \ldots \oplus \tilde{\boldsymbol{x}}_w^\top}_{k+1}\big]^\top,$$

where $\tilde{\boldsymbol{x}}_w = [\text{ReLU}(\boldsymbol{x}_w^\top) \oplus \text{ReLU}(-\boldsymbol{x}_w^\top)]^\top$. Observe that this encoder can be parametrized using a 2-layer MLP, and that $\tilde{\boldsymbol{x}}_w$ can be decoded using a single linear layer to get back to the initial features:

$$\text{DEC}(\tilde{\boldsymbol{x}}_w) = \text{DEC}\left(\big[\text{ReLU}(\boldsymbol{x}_w^\top) \oplus \text{ReLU}(-\boldsymbol{x}_w^\top)\big]^\top\right) = \boldsymbol{x}_w$$

**Individualizing the graph.** Importantly, we encode the features using $2(k+1)d^{(0)}$ dimensions in order to be able to preserve the original node features. Using the construction from Lemma A.1, we can ensure that every pair of nodes in the connected graph have different features with probability $1 - \delta_1$. However, if we do this naïvely, then the node features will be changed before we can transmit them to the target node. We therefore make sure that the width of the CO-GNN architecture from Lemma A.1 is increased to $2(k+1)d^{(0)}$ dimensions such that it applies the identity mapping on all features beyond the first $2d^{(0)}$ components. This way we make sure that all feature components beyond the first $2d^{(0)}$ components are preserved. The existence of such a CO-GNN is straightforward since we can always do an identity mapping using base environment models such SUMGNNs. We use $L_1$ CO-GNN layers for this part of the construction.

In order for our architecture to retain a positive representation for all nodes, we now construct 2 additional layers which encode the representation $\boldsymbol{h}_w^{(L)} \in \mathbb{R}^{2(k+1)d^{(0)}}$ of each node $w$ as follows:

$$[\text{ReLU}(\boldsymbol{q}_w^\top) \oplus \text{ReLU}(-\boldsymbol{q}_w^\top) \oplus \tilde{\boldsymbol{x}}_w^\top \oplus \ldots \oplus \tilde{\boldsymbol{x}}_w^\top]^\top$$

where $\boldsymbol{q}_w \in \mathbb{R}^{2d^{(0)}}$ denotes a vector of the first $2d^{(0)}$ entries of $\boldsymbol{h}_w^{(L_1)}$.

**Transmitting information.** Consider a shortest path $u_1 = w_0 \to w_1 \to \cdots \to w_r \to w_{r+1} = v$ of length $r_1$ from node $u_1$ to node $v$. We use exactly $r_1$ CO-GNN layers in this part of the construction. For the first these layers, the action network assigns the following actions to these nodes:

- $w_0$ performs the action BROADCAST,

- $w_1$ performs the action LISTEN, and

- all other nodes are perform the action ISOLATE.

This is then repeated in the remaining layers, for all consecutive pairs $w_i, w_{i+1}, 0 \le i \le r$ until the whole path is traversed. That is, at every layer, all graph edges are removed except the one between $w_i$ and $w_{i+1}$, for each $0 \le i \le r$. By construction each element in the node representations is positive and so we can ignore the ReLU.

We apply the former construction such that it acts on entries $2d^{(0)}$ to $3d^{(0)}$ of the node representations, resulting in the following representation for node $v$:

$$[\text{ReLU}(\boldsymbol{q}_w^\top) \oplus \text{ReLU}(-\boldsymbol{q}_w^\top) \oplus \tilde{\boldsymbol{x}}_{u_1} \oplus \tilde{\boldsymbol{x}}_v \oplus \ldots \oplus \tilde{\boldsymbol{x}}_v]$$

where $\boldsymbol{q}_w \in \mathbb{R}^{2d^{(0)}}$ denotes a vector of the first $2d^{(0)}$ entries of $\boldsymbol{h}_w^{(L_1)}$.

We denote the probability in which node $y$ does not follow the construction at stage $1 \le t \le r$ by $\beta_y^{(t)}$ such that the probability that all graph edges are removed except the one between $w_i$ and $w_{i+1}$ at stage $t$ is lower bounded by $(1 - \beta)^{|V|}$, where $\beta = \max_{y \in V}(\beta_y)$. Thus, the probability that the construction holds is bounded by $(1 - \beta)^{|V|r_1}$.

The same process is then repeated for nodes $u_i$, $2 \le i \le k$, acting on the entries $(k+1)d^{(0)}$ to $(k+2)d^{(0)}$ of the node representations and resulting in the following representation for node $v$:

$$[\text{ReLU}(\boldsymbol{q}_w^\top) \oplus \text{ReLU}(-\boldsymbol{q}_w^\top) \oplus \tilde{\boldsymbol{x}}_{u_1} \oplus \tilde{\boldsymbol{x}}_{u_2} \oplus \ldots \oplus \tilde{\boldsymbol{x}}_{u_k}]$$

In order to decode the positive features, we construct the feature decoder $\text{DEC}' : \mathbb{R}^{2(k+2)d^{(0)}} \to \mathbb{R}^{(k+1)d^{(0)}}$, that for $1 \le i \le k$ applies DEC to entries $2(i+1)d^{(0)}$ to $(i+2)d^{(0)}$ of its input as follows:

$$[\text{DEC}(\tilde{\boldsymbol{x}}_{u_1}) \oplus \ldots \oplus \text{DEC}(\tilde{\boldsymbol{x}}_{u_k})] = [\boldsymbol{x}_{u_1} \oplus \ldots \oplus \boldsymbol{x}_{u_k}]$$

Given $\epsilon, \delta$, we set:

$$\delta_2 = 1 - \frac{1 - \delta}{(1 - \delta_1)(1 - \beta)^{|V| \sum_{i=1}^{k+1} r_i}} > 0.$$

Having transmitted and decoded all the required features into $\boldsymbol{x} = [\boldsymbol{x}_1 \oplus \ldots \oplus \boldsymbol{x}_k]$, where $\boldsymbol{x}_i$ denotes the vector of entries $id^{(0)}$ to $(i+1)d^{(0)}$ for $0 \leq i \leq k$, we can now use an MLP : $\mathbb{R}^{(k+1)d^{(0)}} \to \mathbb{R}^d$ and the universal approximation property to map this vector to the final representation $\boldsymbol{h}_v^{(L)}$ such that:

$$\mathbb{P}(|\boldsymbol{h}_v^{(L)} - f(\boldsymbol{x}_{u_1}, \ldots \boldsymbol{x}_{u_k})| < \epsilon) \geq (1 - \delta_1)(1 - \beta)^{|V| \sum_{i=1}^{k+1} r_i}(1 - \delta_2)$$
$$\geq 1 - \delta.$$

The construction hence requires $\left(L = L_1 + 2 + \sum_{i=0}^{k} r_i\right)$ CO-GNN layers. □

## B  ANALYZING THE ACTIONS

The action mechanism used by CO-GNNs is a key component in the new message-passing paradigm. However, as the action distribution, sampling, and aggregation are performed in a black-box manner (no access to them from outside of the architecture), it is hard to understand the correlation between them and CO-GNNs favorable empirical results. The purpose of this section is to open this black box and to provide additional insights regarding the actions being learned by CO-GNNs.

An obvious idea would be to inspect the learned action distributions on different datasets, but the action distributions alone may not provide a clear picture of the topological structure of the graph. For example, two connected nodes that choose to ISOLATE achieve the same topology as nodes that choose both to BROADCAST or LISTEN. This is a simple consequence of the immense number of action configurations and their rich set of interactions.

In an effort to better understand the learned graph topology, we can instead inspect directly the induced directed graphs at every layer. To quantify the evolution of the graphs across layers, we present the *ratio of the directed edges that are kept* across the different layers in Figure 7. We record the directed edge ratio over the 10 different layers of our best, fully trained 10 CO-GNN$(\mu, \mu)$ model on the roman-empire (Platonov et al., 2023) and cora datasets (Pei et al., 2020). We follow the 10 data splits and the methodology of Platonov et al. (2023) and Yang et al. (2016), respectively.

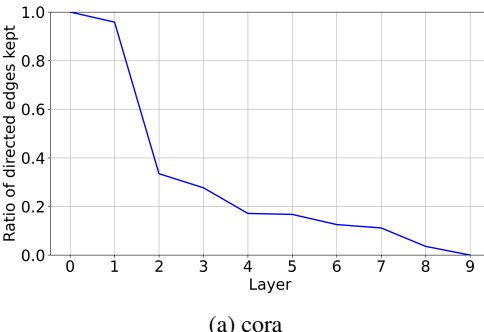
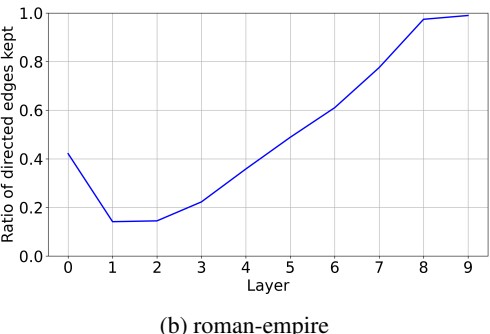

(a) cora                    (b) roman-empire

Figure 7: The ratio of directed edges that are kept on (a) cora (as a homophilic dataset) and on (b) roman-empire (as a heterophilic dataset) for each layer $0 \leq \ell < 10$.

**Dataset and task-specific**: We argued that CO-GNNs are adaptive to different tasks and datasets while discussing their properties in Section 5. This experiment serves as a strong evidence for this statement. Indeed, by inspecting Figure 7, we observe completely *opposite trends* between the two datasets in terms of the evolution of the ratio of edges that are kept across layers:

- On the homophilic dataset cora, the ratio of edges that are kept gradually *decreases* as we go to the deeper layers. In fact, $100\%$ of the edges are kept at layer $\ell = 0$ while *all* edges are dropped at layer $\ell = 9$. This is very insightful because homophilic datasets are known to *not* benefit from using many layers, and the trained CO-GNN model recognizes this by eventually isolating all the nodes. This is particularly the case for cora, where classical MPNNs typically achieve their best performance with 1-3 layers.

- On the heterophilic dataset roman-empire, the ratio of edges that are kept gradually *increases* after $\ell = 1$ as we go to the deeper layers. Initially, $\sim 42\%$ of the edges are kept at layer $\ell = 0$

while eventually this reaches 99% at layer $\ell = 9$. This is interesting, since in heterophilic graphs, edges tend to connect nodes of different classes and so classical MPNNS, which aggregate information based on the homophily assumption perform poorly. Although, CO-GNN model uses these classical models it compensates by intricately controlling information flow. It seems that CO-GNN model manages to capture the heterophilous aspect of the dataset by restricting the flow of information in the early layer and slowly enabling it the deeper the layer (the further away the nodes), which might be a great benefit to its success over heterophilic benchmarks.

## C  RUNTIME ANALYSIS

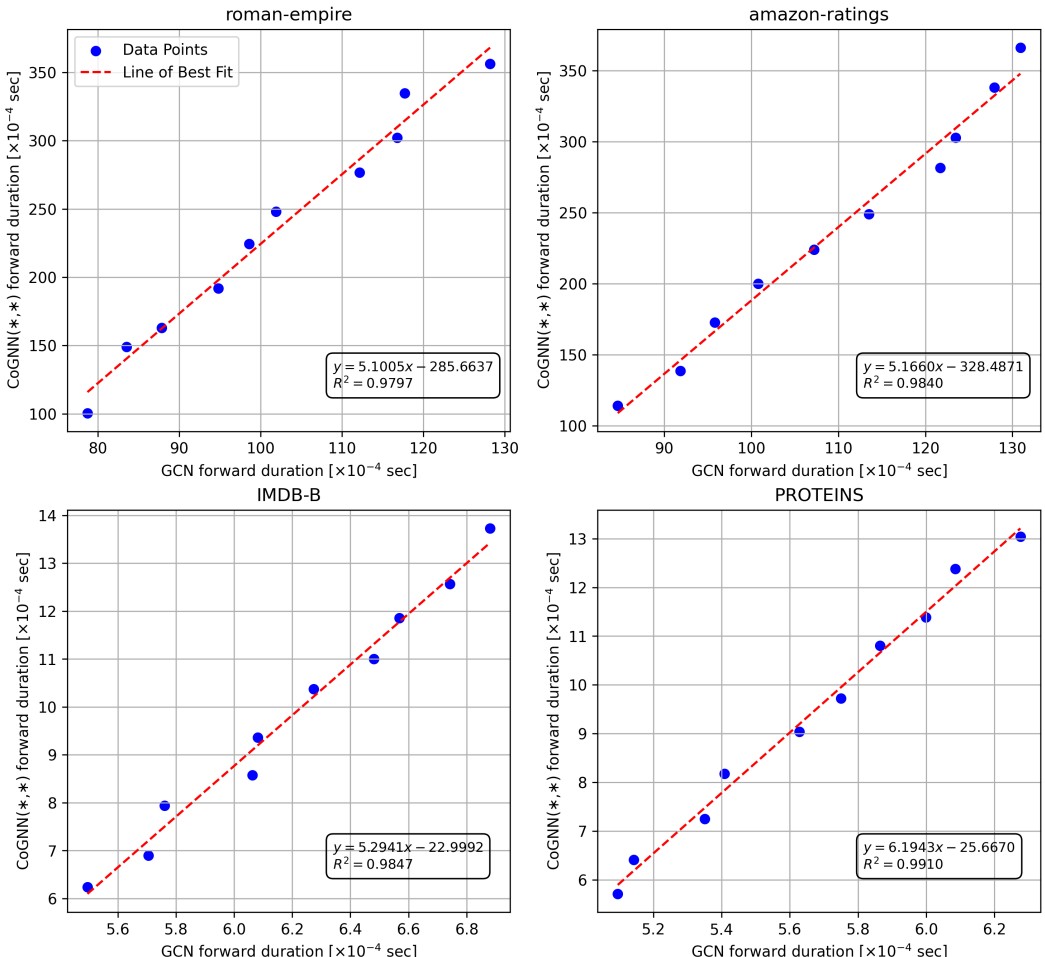

Figure 8: Empirical runtimes: CO-GNN$(*, *)$ forward pass duration as a function of GCN forward pass duration on $4$ datasets.

Consider a GCN model with $L$ layers and a hidden dimension of $d$ on an input graph $G = (V, E, \boldsymbol{X})$. Wu et al. (2019) has shown the time complexity of this model to be $\mathcal{O}(Ld(|E|d + |V|))$. To extend this analysis to CO-GNNs, let us consider a CO-GNN$(*, *)$ architecture composed of:

- a GCN environment network $\eta$ with $L_\eta$ layers and hidden dimension of $d_\eta$, and
- a GCN action network $\pi$ with $L_\pi$ layers and hidden dimension of $d_\pi$.

A single CO-GNN layer first computes the actions for each node by feeding node representations through the action network $\pi$, which is then used in the aggregation performed by the environment layer. This means that the time complexity of a single CO-GNN layer is

$\mathcal{O}(L_\pi d_\pi(|E|d_\pi + |V|) + d_\eta(|E|d_\eta + |V|))$. The time complexity of the whole CO-GNN architecture is then $\mathcal{O}(L_\eta L_\pi d_\pi(|E|d_\pi + |V|) + L_\eta d_\eta(|E|d_\eta + |V|))$.

Typically, the hidden dimensions of the environment network and action network match. In all of our experiments, the depth of the action network $L_\pi$ is much smaller (typically $\leq 3$) than that of the environment network $L_\eta$. Therefore, assuming $L_\pi << L_\eta$ we get that a runtime complexity of $\mathcal{O}(L_\eta d_\eta(|E|d_\eta + |V|))$, matching the runtime of a GCN model.

To empirically confirm the efficiency of CO-GNNs, we report in Figure 8 the duration of a forward pass of a CO-GNN$(*, *)$ and GCN with matching hyperparameters across multiple datasets. From Figure 8, it is evident that the increase in runtime is linearly related to its corresponding base model with $R^2$ values higher or equal to 0.98 across 4 datasets from different domains. Note that, for the datasets IMDB-B and PROTEINS, we report the average forward duration for a single graph in a batch.

## D  ADDITIONAL EXPERIMENTS

### D.1  EXPRESSIVITY EXPERIMENT

In Proposition 5.1 we state that CO-GNNs can distinguish between pairs of graphs which are 1-WL indistinguishable. We validate this with a simple synthetic dataset: CYCLES. Similar to the pair of graphs presented in Figure 5, CYCLES consists of 7 pairs of undirected graphs, where the first graph is a $k$-cycle for $k \in [6, 12]$ and the second graph is a disjoint union of a $(k-3)$-cycle and a triangle. The task is to correctly identify the cycle graphs. As the pairs are 1-WL indistinguishable, solving this task implies a strictly higher expressive power than 1-WL. CO-GNN$(\Sigma, \Sigma)$ and CO-GNN$(\mu, \mu)$ achieve $100\%$ training accuracy, perfectly classifying the cycles, whereas their corresponding classical MPNNs SUMGNN and MEANGNN achieve a random guess accuracy of $50\%$. These results imply that CO-GNN can increase the expressive power of their classical counterparts.

### D.2  LONG-RANGE INTERACTIONS

To validate the performance of CO-GNNs on long-range tasks, we experiment with the recent LRGB benchmark (Dwivedi et al., 2022).

**Setup.** We train CO-GNN$(*, *)$ and CO-GNN$(\epsilon, \epsilon)$ CO-GNN$(\epsilon, \epsilon)$ on LRGB and report the unweighted mean Average Precision (AP) for Peptides-func. All experiments are run 4 times with 4 different seeds and follow the data splits provided by Dwivedi et al. (2022). Following the methodology in Tönshoff et al. (2023), we used AdamW as optimizer and cosine-with-warmup scheduler. We also use the provided results for GCN (Kipf & Welling, 2017), GCNII (Chen et al., 2020), GINE (Xu et al., 2019), GatedGCN (Bresson & Laurent, 2018), CRaWl (Tönshoff et al., 2023), DRew (Gutteridge et al., 2023), Exphormer (Shirzad et al., 2023), GRIT (Ma et al., 2023), Graph-ViT / G-MLPMixer (He et al., 2023).

Table 4: Results on LRGB. Top three models are colored by First, Second, Third.

|  | Peptides-func |
| --- | --- |
| GCN | $0.6860 \pm 0.0050$ |
| GINE | $0.6621 \pm 0.0067$ |
| GatedGCN | $0.6765 \pm 0.0047$ |
| CRaWl | $0.7074 \pm 0.0032$ |
| DRew | $0.7150 \pm 0.0044$ |
| Exphormer | $0.6527 \pm 0.0043$ |
| GRIT | $0.6988 \pm 0.0082$ |
| Graph-ViT | $0.6942 \pm 0.0075$ |
| G-MLPMixer | $0.6921 \pm 0.0054$ |
| CO-GNN$(*, *)$ | $0.6990 \pm 0.0093$ |
| CO-GNN$(\epsilon, \epsilon)$ | $0.6963 \pm 0.0076$ |

**Results.** We follow Tönshoff et al. (2023) who identified that the previously reported large performance gaps between classical MPNNs and transformer-based models can be closed by a more extensive tuning of MPNNs. In light of this, we note that the performance gap between different models is not large. Classical MPNNs such as GCN, GINE, and GatedGCN surpass some transformer-based approaches such as Exphormer. CO-GNN$(*, *)$ further improves on the competitive GCN and is the third best performing model after DRew and CRaWl. Similarly, CO-GNN$(\epsilon, \epsilon)$ closely matches CO-GNN$(*, *)$ and is substantially better than its base architecture GIN. This experiment further suggests that exploring different classes of CO-GNNs is a promising direction, as CO-GNNs typically boost the performance of their underlying base architecture.

# E  FURTHER DETAILS OF THE EXPERIMENTS REPORTED IN THE PAPER

In this section, we present the Gumbel distribution and the Gumbel-softmax temperature, the data generation for ROOTNEIGHBORS, dataset statistics, and the hyperparameters.

## E.1  THE GUMBEL DISTRIBUTION AND THE GUMBEL-SOFTMAX TEMPERATURE

The Gumbel distribution is used to model the distribution of the maximum (or the minimum) of a set of random variables. Its probability density function has a distinctive, skewed shape, with heavy tails, making it a valuable tool for analyzing and quantifying the likelihood of rare and extreme occurrences. By applying the Gumbel distribution to the logits or scores associated with discrete choices, the Gumbel-Softmax estimator transforms them into a probability distribution over the discrete options. The probability density function of a variable that follows $X \sim \text{Gumbel}(0, 1)$ is the following:

$$f(x) = e^{-x + e^{-x}}$$

We also visualize the probability density function of $\text{Gumbel}(0, 1)$ in Figure 9.

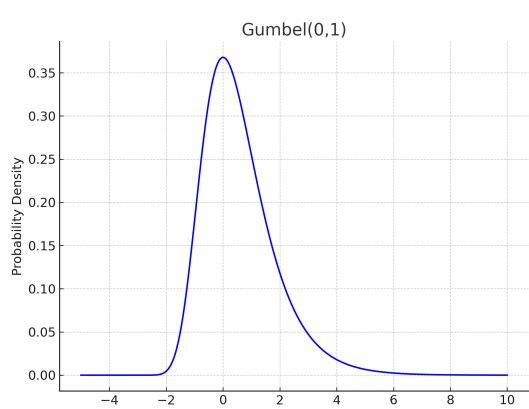

Figure 9: The pdf $f(x) = e^{-x+e^{-x}}$ of $\text{Gumbel}(0, 1)$.

The Straight-through Gumbel-softmax estimator is known to benefit from learning an inverse-temperature before sampling an action, which we use in our experimental setup. For a given graph $G = (V, E, \boldsymbol{X})$ the inverse-temperature of node $v \in V$ is estimated by applying a bias-free linear layer $L : \mathbb{R}^d \to \mathbb{R}$ to the intermediate representation $\boldsymbol{h} \in \mathbb{R}^d$. To ensure the temperature is positive, an approximation of the ReLU function with a bias hyperparameter $\tau \in \mathbb{R}$ is subsequently applied:

$$\frac{1}{\tau(\boldsymbol{h})} = \log\left(1 + \exp\left(\omega^T \boldsymbol{h}\right)\right) + \tau_0$$

where $\tau_0$ controls the maximum possible value for the temperature.

## E.2  ROOTNEIGHBORS: DATASET GENERATION

In Section 6.1, we compare CO-GNNs to a class of MPNNs on a dedicated synthetic dataset ROOTNEIGHBORS in order to assess the model's ability to redirect the information flow. ROOTNEIGHBORS consists of 3000 trees of depth 2 with random node features of dimension $d = 5$ which is generated as follows:

- **Features**: Each feature is independently sampled from a uniform distribution $U[-2, 2]$s.
- **Level-1 nodes**: The number of nodes in the first level of each tree in the train, validation, and test set is sampled from a uniform distribution $U[3, 10]$, $U[5, 12]$, and $U[5, 12]$ respectively. Then, the degrees of the level-1 nodes are sampled as follows:
  - The number of level-1 nodes with a degree of 6 is sampled independently from a uniform distribution $U[1, 3]$, $U[3, 5]$, $U[3, 5]$ for the train, validation, and test set, respectively.
  - The degree of the remaining level-1 nodes are sampled from the uniform distribution $U[2, 3]$.

We use a train, validation, and test split of equal size.

## E.3  DATASET STATISTICS

The statistics of the real-world long-range, node-based, and graph-based benchmarks used can be found in Tables 5 to 7.

Table 5: Statistics of the long-range graph benchmarks (LRGB).

|  | Peptides-func |
|---|---|
| # graphs | 15535 |
| # average nodes | 150.94 |
| # average edges | 307.30 |
| # classes | 10 |
| metrics | AP |

Table 6: Statistics of the node classification benchmarks.

|  | roman-empire | amazon-ratings | minesweeper | tolokers | questions |
|---|---|---|---|---|---|
| # nodes | 22662 | 24492 | 10000 | 11758 | 48921 |
| # edges | 32927 | 93050 | 39402 | 519000 | 153540 |
| # node features | 300 | 300 | 7 | 10 | 301 |
| # classes | 18 | 5 | 2 | 2 | 2 |
| edge homophily | 0.05 | 0.38 | 0.68 | 0.59 | 0.84 |
| metrics | ACC | ACC | AUC-ROC | AUC-ROC | AUC-ROC |

Table 7: Statistics of the graph classification benchmarks.

|  | IMDB-B | IMDB-M | REDDIT-B | NCI1 | PROTEINS | ENZYMES |
|---|---|---|---|---|---|---|
| # graphs | 1000 | 1500 | 2000 | 4110 | 1113 | 600 |
| # average nodes | 19.77 | 13.00 | 429.63 | 29.87 | 39.06 | 32.63 |
| # average edges | 96.53 | 65.94 | 497.75 | 32.30 | 72.82 | 64.14 |
| # classes | 2 | 3 | 2 | 2 | 2 | 6 |
| metrics | ACC | ACC | ACC | ACC | ACC | ACC |

## E.4 HYPERPARAMETERS

In Tables 8 to 11, we report the hyper-parameters used in our synthetic experiments and real-world long-range, node-based and graph-based benchmarks.

Table 8: Hyper-parameters used for ROOTNEIGHBORS and CYCLES.

|  | ROOTNEIGHBORS | CYCLES |
|---|---|---|
| $\eta$ # layers | 1 | 10, 20, 30 |
| $\eta$ dim | 16, 32 | 8, 16, 32, 64 |
| $\pi$ # layers | 1, 2 | 1, 2 |
| $\pi$ dim | 8, 16 | 16 |
| learned temp | ✓ | - |
| temp | - | 1 |
| $\tau_0$ | 0.1 | - |
| # epochs | 10000 | 1000 |
| dropout | 0 | 0 |
| learn rate | $10^{-3}$ | $10^{-3}, 10^{-4}$ |
| batch size | - | 14 |

Table 9: Hyper-parameters used for the long-range graph benchmarks (LRGB).

|  | Peptides-func |
|---|---|
| $\eta$ # layers | 5-9 |
| $\eta$ dim | 200,300 |
| $\pi$ # layers | 1-3 |
| $\pi$ dim | 8,16,32 |
| learned temp | ✓ |
| $\tau_0$ | 0.5 |
| # epochs | 500 |
| dropout | 0 |
| learn rate | $3 \cdot 10^{-4}, 10^{-3}$ |
| # decoder layer | 2,3 |
| # warmup epochs | 5 |
| positional encoding | LapPE, RWSE |
| batch norm | ✓ |
| skip connections | ✓ |

Table 10: Hyper-parameters used for the node classification benchmarks.

|  | roman-empire | amazon-ratings | minesweeper | tolokers | questions |
|---|---|---|---|---|---|
| $\eta$ # layers | 5-12 | 5-10 | 8-15 | 5-10 | 5-9 |
| $\eta$ dim | 128,256,512 | 128,256 | 32,64,128 | 16,32 | 32,64 |
| $\pi$ # layers | 1-3 | 1-6 | 1-3 | 1-3 | 1-3 |
| $\pi$ dim | 4,8,16 | 4,8,16,32 | 4,8,16,32,64 | 4,8,16,32 | 4,8,16,32 |
| learned temp | ✓ | ✓ | ✓ | ✓ | ✓ |
| $\tau_0$ | 0,0.1 | 0,0.1 | 0,0.1 | 0,0.1 | 0,0.1 |
| # epochs | 3000 | 3000 | 3000 | 3000 | 3000 |
| dropout | 0.2 | 0.2 | 0.2 | 0.2 | 0.2 |
| learn rate | $3 \cdot 10^{-3}, 3 \cdot 10^{-5}$ | $3 \cdot 10^{-4}, 3 \cdot 10^{-5}$ | $3 \cdot 10^{-3}, 3 \cdot 10^{-5}$ | $3 \cdot 10^{-3}$ | $10^{-3}, 10^{-2}$ |
| activation function | GeLU | GeLU | GeLU | GeLU | GeLU |
| skip connections | ✓ | ✓ | ✓ | ✓ | ✓ |
| layer normalization | ✓ | ✓ | ✓ | ✓ | ✓ |

Table 11: Hyper-parameters used for social networks and proteins datasets.

|  | IMDB-B | IMDB-M | REDDIT-B | NCI1 | PROTEINS | ENZYMES |
|---|---|---|---|---|---|---|
| $\eta$ # layers | 1 | 1 | 3,6 | 2,5 | 3,5 | 1,2 |
| $\eta$ dim | 32,64 | 64,256 | 128,256 | 64,128,256 | 64 | 128,256 |
| $\pi$ # layers | 2 | 3 | 1,2 | 2 | 1,2 | 1 |
| $\pi$ dim | 16,32 | 16 | 16,32 | 8, 16 | 8 | 8 |
| learned temp. | ✓ | ✓ | ✓ | ✓ | ✓ | ✓ |
| $\tau_0$ | 0.1 | 0.1 | 0.1 | 0.5 | 0.5 | 0.5 |
| # epochs | 5000 | 5000 | 5000 | 3000 | 3000 | 3000 |
| dropout | 0.5 | 0.5 | 0.5 | 0 | 0 | 0 |
| learn rate | $10^{-4}$ | $10^{-3}$ | $10^{-3}$ | $10^{-3}, 10^{-2}$ | $10^{-3}$ | $10^{-3}$ |
| pooling | mean | mean | mean | mean | mean | mean |
| batch size | 32 | 32 | 32 | 32 | 32 | 32 |
| scheduler step size | 50 | 50 | 50 | 50 | 50 | 50 |
| scheduler gamma | 0.5 | 0.5 | 0.5 | 0.5 | 0.5 | 0.5 |

