# Rebuttal for Cooperative Graph Neural Networks

## 1 Experiments on homophilic node classification

In this experiment, we evaluate Co-GNNs on the homophilic node classification benchmarks cora and pubmed (Sen et al., 2008).

**Setup.** We assess MeanGNN, SumGNN and their corresponding Co-GNNs counterparts Co-GNN($\mu, \mu$) and Co-GNN($\Sigma, \Sigma$) on the homophilic graphs and their 10 fixed splits provided by Pei et al. (2020), where we report the mean accuracy, standard deviation and the accuracy gain due to the application of Co-GNN. We also use the results provided by Bodnar et al. (2023) for the classical baseline: GCN (Kipf & Welling, 2017), GraphSAGE (Hamilton et al., 2017), GAT (Veličković et al., 2018), Geom-GCN (Pei et al., 2020) and GCNII (Chen et al., 2020).

**Results.** Table 1 illustrates a modest performance increase of 1-2% across all datasets when transitioning from SumGNN, MeanGNN, and GCN to their respective Co-GNN counterparts. These datasets are highly homophilic, but Co-GNNs nonetheless show improvements on these datasets (even though, modest) compared to their environment/action network architectures, leading to competitive results overall.

Table 1: Results on homophilic datasets. Top three models are colored by First, Second, Third.

|  | pubmed | cora |
|---|---|---|
| MLP | $87.16 \pm 0.37$ | $75.69 \pm 2.00$ |
| GCN | $88.42 \pm 0.50$ | $86.98 \pm 1.27$ |
| GraphSAGE | $88.45 \pm 0.50$ | $86.90 \pm 1.04$ |
| GAT | $87.30 \pm 1.10$ | $86.33 \pm 0.48$ |
| Geom-GCN | $87.53 \pm 0.44$ | $85.35 \pm 1.57$ |
| GCNII | $90.15 \pm 0.43$ | $88.37 \pm 1.25$ |
| SumGNN | $88.58 \pm 0.57$ | $84.80 \pm 1.71$ |
| MeanGNN | $88.66 \pm 0.44$ | $84.50 \pm 1.25$ |
| Co-GNN($\Sigma, \Sigma$) | $89.39 \pm 0.39$ | $86.43 \pm 1.28$ |
| Co-GNN($\mu, \mu$) | $89.60 \pm 0.42$ | $86.53 \pm 1.20$ |
| Co-GNN($*, *$) | $89.51 \pm 0.88$ | $87.44 \pm 0.85$ |
| SumGNN gain | +0.91% | +1.92% |
| MeanGNN gain | +1.06% | +2.40% |
| GCN gain | +1.25% | +0.53% |

## 2 Experiments on ZINC

In this experiment, we evaluate Co-GNNs on the ZINC (12k graphs) graph classification benchmark (Dwivedi et al., 2023).

**Setup.** We evaluate SumGNN and its Co-GNN counterpart Co-GNN($\mu, \mu$) on the ZINC (12k graphs) dataset with no edge features, and report the Mean Average Error (MAE) and standard deviation over 10 different seeds. The results for the baselines GCN (Kipf & Welling, 2017), GIN (Xu et al., 2019), GraphSAGE (Hamilton et al., 2017), GAT (Veličković et al., 2018), MoNet (Monti et al., 2017), GatedGCN (Bresson & Laurent, 2018) and PNA (Corso et al., 2020) are from Bouritsas et al. (2022). We also the accuracy gain due to the application of Co-GNNs.

**Results.** The objective of this task strongly correlates with cycle counts (rings in organic molecules), so those subgraph GNNs that explicitly inject such information will perform strongly. For fairness, we only considered GNN architectures that do not inject such subgraph counts. From Table 2, we can see that Co-GNNs outperform all architectures, including GAT, GatedGCN, and PNA.

Table 2: Results on ZINC(12k). Top three models are colored by First, Second, Third.

|  | ZINC |
|---|---|
| GCN | $0.469 \pm 0.002$ |
| GIN | $0.408 \pm 0.008$ |
| GraphSAGE | $0.410 \pm 0.005$ |
| GAT | $0.463 \pm 0.002$ |
| MoNet | $0.407 \pm 0.007$ |
| GatedGCN | $0.422 \pm 0.006$ |
| PNA | $0.320 \pm 0.032$ |
| SumGNN | $0.464 \pm 0.005$ |
| Co-GNN($\Sigma, \Sigma$) | $0.316 \pm 0.010$ |
| SumGNN gain | -31.90% |

## 3 ABLATION STUDY ON HETEROPHILIC DATASETS

In this experiment, we perform an ablation study for Co-GNN architectures on the heterophilic graph classification benchmarks roman-empire, amazon-ratings, minesweeper, tolokers, and questions (Platonov et al., 2023).

Table 3: Ablation study on heterophilic datasets.

|                              | roman-empire      | amazon-ratings    | minesweeper       | tolokers          | questions         |
| ---------------------------- | ----------------- | ----------------- | ----------------- | ----------------- | ----------------- |
| SUMGNN                       | $82.82 \pm 0.47$  | $50.50 \pm 0.47$  | $90.81 \pm 0.64$  | $81.73 \pm 0.86$  | $78.88 \pm 1.10$  |
| MEANGNN                      | $85.05 \pm 0.72$  | $53.08 \pm 0.83$  | $92.98 \pm 0.48$  | $81.79 \pm 0.55$  | $77.05 \pm 1.10$  |
| Co-GNN$(\Sigma, \Sigma)$     | $91.57 \pm 0.32$  | $51.28 \pm 0.56$  | $95.09 \pm 1.18$  | $83.36 \pm 0.89$  | $80.02 \pm 0.86$  |
| Co-GNN$(\mu, \mu)$           | $91.37 \pm 0.35$  | $54.17 \pm 0.37$  | $97.31 \pm 0.41$  | $84.45 \pm 1.17$  | $76.54 \pm 0.95$  |
| SUMGNN gain                  | +10.56%           | +1.54%            | +4.71%            | +1.99%            | +1.44%            |
| MEANGNN gain                 | +7.43%            | +2.05%            | +4.66%            | +3.25%            | -0.66%            |

**Setup.** We evaluate MEANGNN, SUMGNN on the 5 heterophilic graphs, following the 10 data splits and the methodology of Platonov et al. (2023). We report the accuracy, standard deviation and accuracy gain due to the application of Co-GNNs for roman-empire and amazon-ratings. We also report the mean ROC AUC, standard deviation and ROC AUC gain due to the application of Co-GNNs for minesweeper, tolokers, and questions.

**Results.** Table 3 suggest a clear trend leading to improvements as a result of using Co-GNNs. The increase in evaluation metrics is particularly prominent on the roman-empire (average increase of 9.00% accuracy) and minesweeper datasets (average increase of 4.69% ROC AUC), further showing that Co-GNNs effectiveness is correlated with the graph topology and the task.

## 4 EXPERIMENTS ON REDDIT-M

In this experiment, we evaluate Co-GNNs on the REDDIT-M (Morris et al., 2020) graph classification dataset.

**Setup.** We evaluate Co-GNN$(\Sigma, \Sigma)$ and Co-GNN$(\mu, \mu)$ on the REDDIT-M benchmark, following the risk assessment protocol of Errica et al. (2020), and report the mean accuracy and standard deviation. The results for the baselines DGCNN (Wang et al., 2019), DiffPool (Ying et al., 2018), Edge-Conditioned Convolution (ECC) (Simonovsky & Komodakis, 2017), GIN (Xu et al., 2019), GraphSAGE (Hamilton et al., 2017) are from Errica et al. (2020). We also include CGMM (Bacciu et al., 2020), ICGMM$_f$ (Castellana et al., 2022), and GSPN (Errica & Niepert, 2023) as more recent baseline. OOR (Out of Resources) implies extremely long training time or GPU memory usage.

**Results.** Co-GNN models achieve the highest accuracy on REDDIT-M as reported in Table 4, despite using relatively simple action and environment networks.

Table 4: Results on REDDIT-M. Top three models are colored by First, Second, Third.

|                     | REDDIT-M        |
| ------------------- | --------------- |
| DGCNN               | $49.2 \pm 1.2$  |
| DiffPool            | $53.8 \pm 1.4$  |
| ECC                 | OOR             |
| GIN                 | $56.1 \pm 1.7$  |
| GraphSAGE           | $50.0 \pm 1.3$  |
| CGMM                | $52.4 \pm 2.2$  |
| ICGMM$_f$           | $55.6 \pm 1.7$  |
| GSPN                | $55.3 \pm 2.0$  |
| Co-GNN$(\mu, \mu)$  | $56.3 \pm 2.1$  |

## 5 GAT EXPERIMENTS ON ROOTNEIGHBORS

In this experiment, we enhance GATs(Veličković et al., 2018) with an action network and evaluate on our synthetic dataset: ROOTNEIGHBORS.

**Setup.** We trained a Co-GNNs architecture which uses GAT (Veličković et al., 2018) as its action network and SUMGNN as its environment network, where $\alpha$ refers to the GAT architecture. We report the Mean Average Error (MAE).

Table 5: Results on ROOTNEIGHBORS. Top three models are colored by First, Second, Third.

| Model | MAE |
|-------|-----|
| Random | 0.474 |
| SUMGNN | 0.370 |
| MEANGNN | 0.329 |
| GAT | 0.442 |
| CO-GNN$(\Sigma, \Sigma)$ | 0.196 |
| CO-GNN$(\Sigma, \mu)$ | 0.079 |
| CO-GNN$(\Sigma, \alpha)$ | 0.085 |
| SUMGNN gain | -47.03% |
| MEANGNN gain | -75.99% |
| GAT gain | -80.77% |

**Results.** In ROOTNEIGHBORS, information needs to be propagated only from degree-6 nodes, and GAT model, unable to detect node degrees, performs poorly. In other words, GAT may not be able to detect which information needs to be filtered (especially structural ones). Our message-passing paradigm generally increases the performance of GATs. To show this, we experiment with CO-GNN$(\Sigma, \alpha)$ over ROOTNEIGHBORS, which results in an 80% decrease in MAE (Table 5), from the initial GAT performance which was basically a random guess. In this case, the action network allows GAT to determine the right topology, and GAT only needs to learn to average of the features.

## 6    VISUALIZING THE ACTIONS

The CO-GNN architecture benefits from the dynamic topology that is created by the action network. To better understand the learned graph topology, we visualize the topology at each layer in a CO-GNN model over the highly regular minesweepers dataset.

**Dataset.** The minesweeper dataset (Platonov et al., 2023) is a synthetic dataset inspired by the popular game Minesweeper. It is a semi-supervised node classification dataset with a regular $100 \times 100$ grid where each node is connected to eight neighboring nodes. Each node has an input feature of one-hot-encoded representations, showing the number of adjacent mines. A randomly chosen 50% of the nodes has an unknown feature, indicated by a separate binary feature. The task is to correctly identify if the querying node is a mine.

**Setup.** We train a 10-layered CO-GNN$(\mu, \mu)$ model and present the evolution of the graph topology from layer $\ell = 1$ to layer $\ell = 8$. We choose a node (black), and at every layer $\ell$, we depict its neighbors up to distance 10. In this visualization, nodes which are mines are shown in red, and other nodes in blue. The features of non-mine nodes (indicating the number of neighboring mines) are shown explicitly whereas the nodes whose features are hidden are labeled with a question mark. For each layer $\ell$, we gray out the nodes whose information cannot reach the black node with the remaining layers available.

**Results.** Interestingly, in the early layers $\ell = 1, 2, 3, 4$, the action network learns to isolate the right section of the black node, similar to how humans would play this game: The bulk of the nodes without neighboring mines (0 labeled nodes) initially do not help in determining whether the black node is a mine or not. Thus, the action network prioritizes the information flowing from the left sections of the grid where more mines are present. We find this very nice and informative: Action network initially focuses mostly on nodes that are more informative for the task. After identifying the most crucial information and propagating this through the network, it then requires this information to also be communicated with the nodes that initially were labeled with 0. This leads to an almost fully connected grid in the later layers $\ell = 7, 8$.

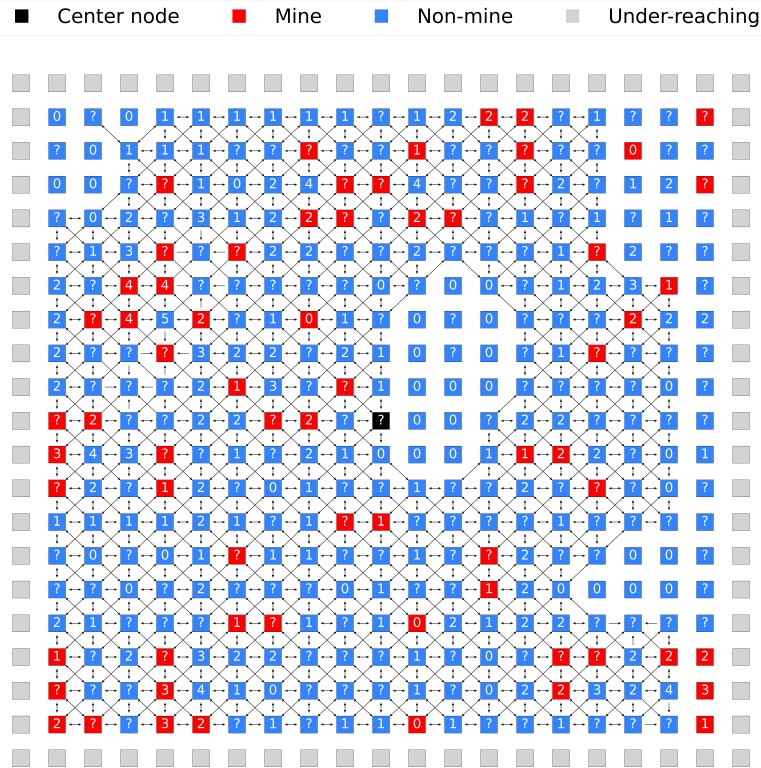

The 10-hop neighborhood at layer $\ell = 1$.

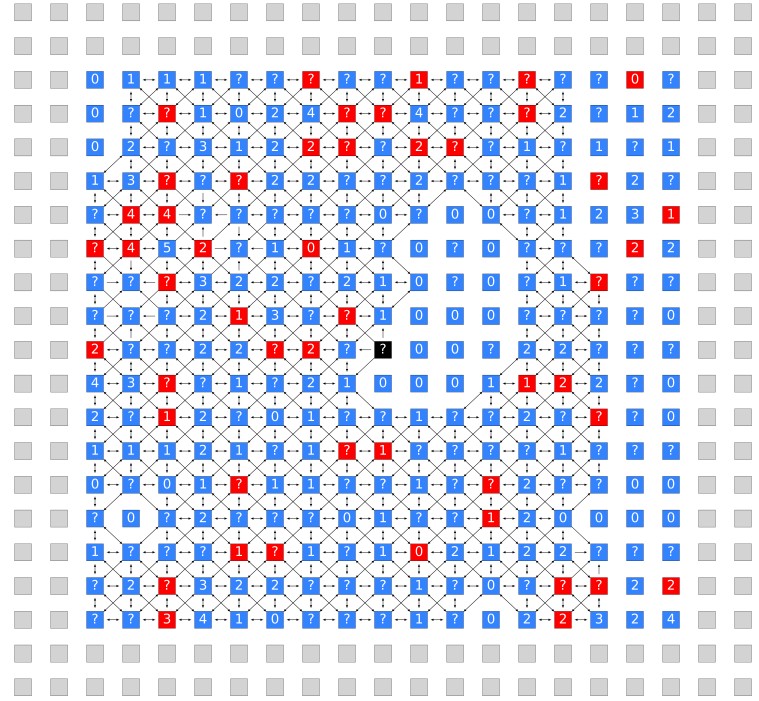

The 10-hop neighborhood at layer $\ell = 2$.

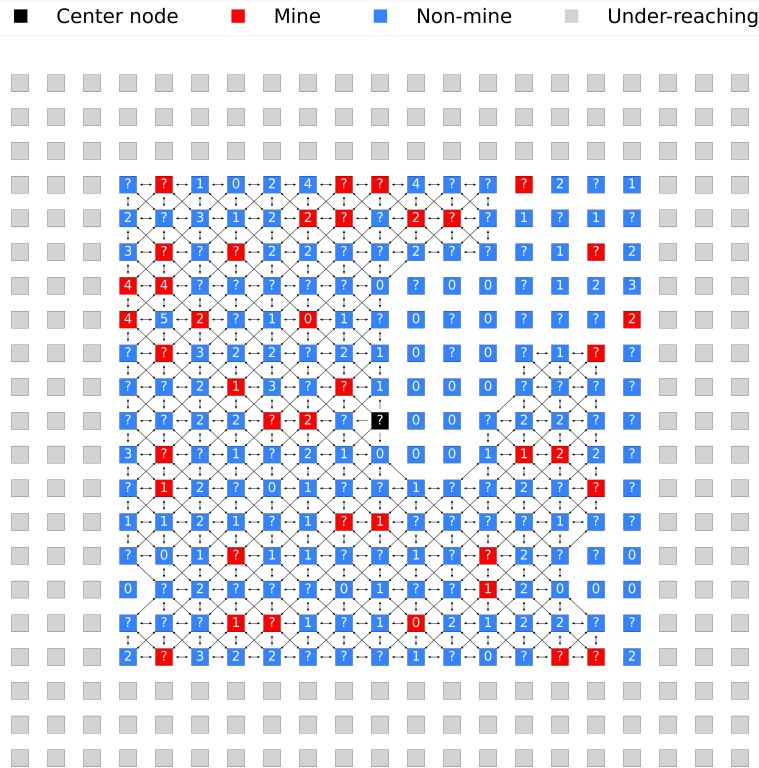

The 10-hop neighborhood at layer $\ell = 3$.

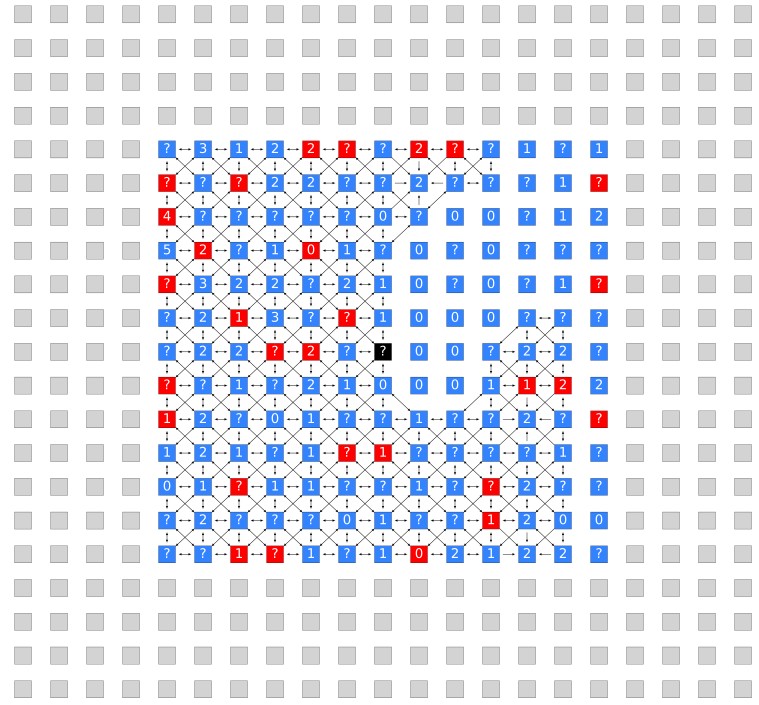

The 10-hop neighborhood at layer $\ell = 4$.

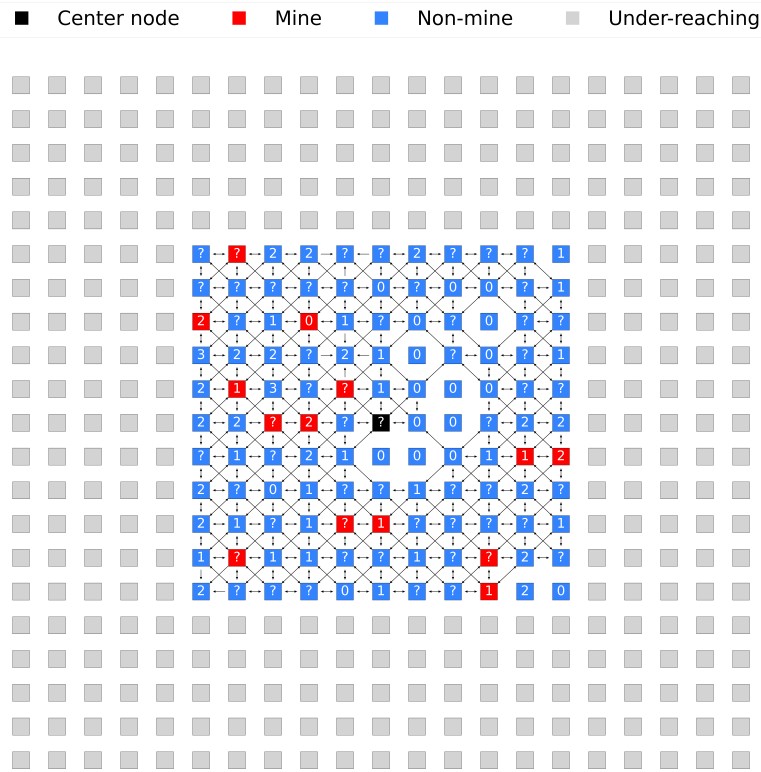

The 10-hop neighborhood at layer $\ell = 5$.

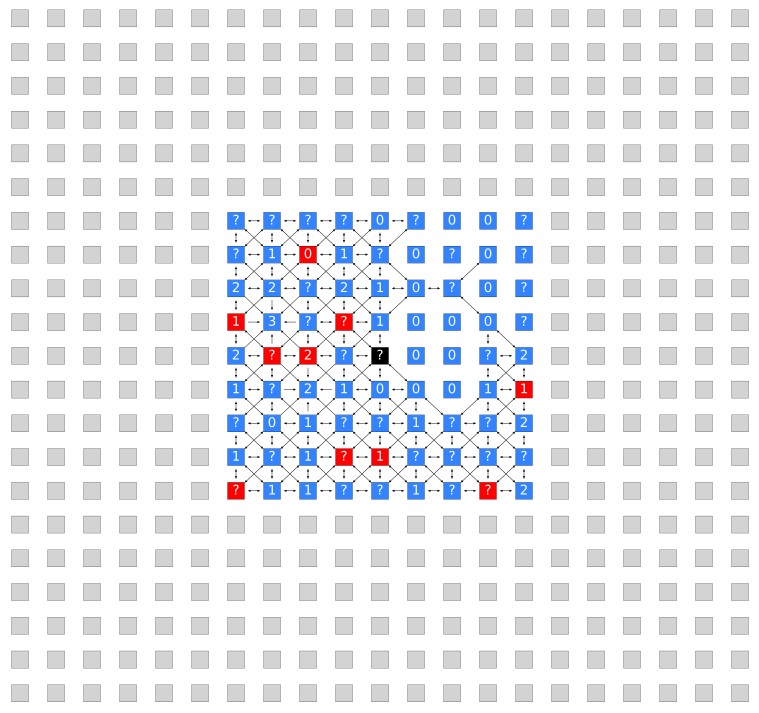

The 10-hop neighborhood at layer $\ell = 6$.

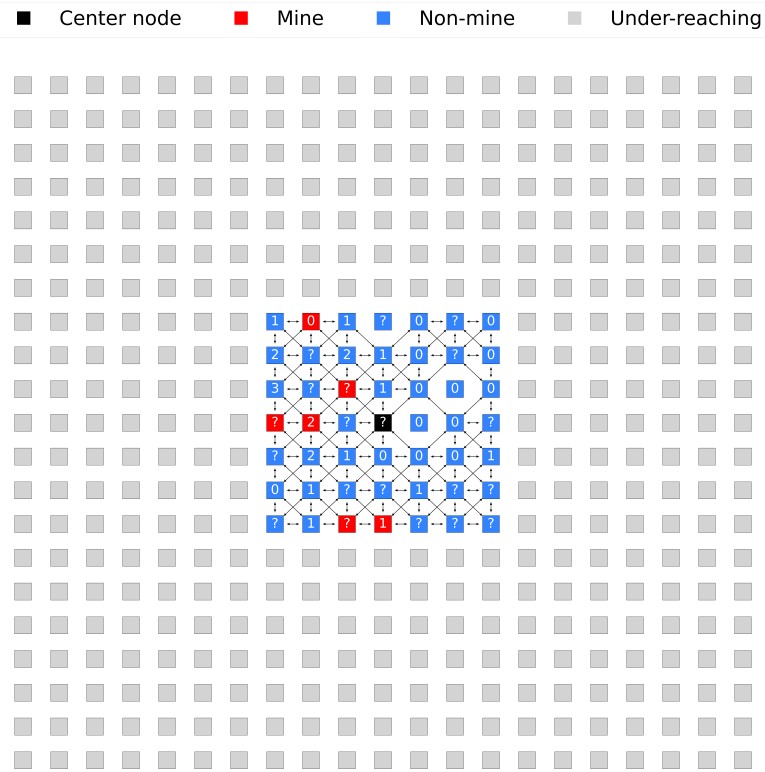

The 10-hop neighborhood at layer $\ell = 7$.

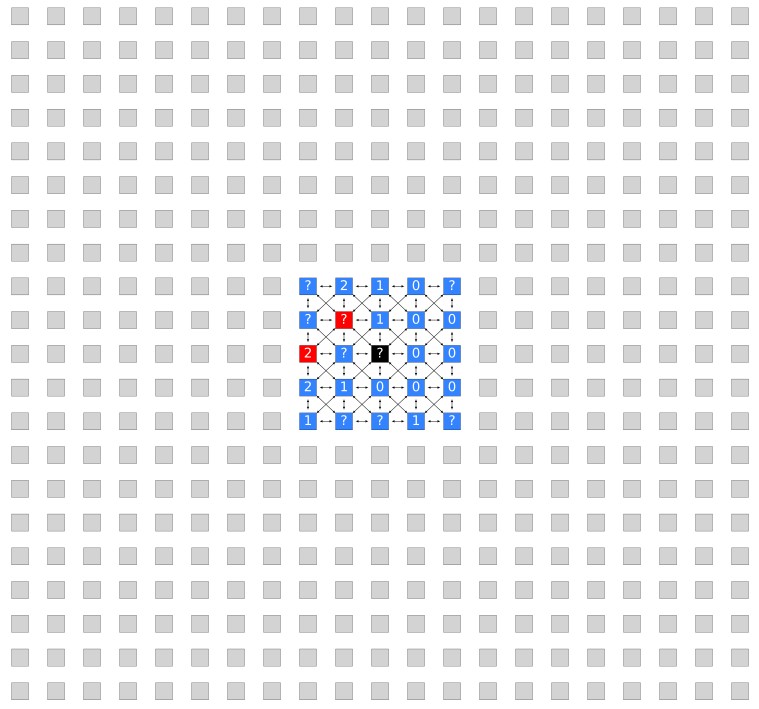

The 10-hop neighborhood at layer $\ell = 8$.