# OpenReview forum: "Cooperative Graph Neural Networks"
_ICLR.cc/2024/Conference — Submitted to ICLR 2024_

### Official Review · Reviewer_9qTL · 2023-10-26

**Soundness:** 2 fair
**Presentation:** 3 good
**Contribution:** 2 fair
**Rating:** 6
**Confidence:** 3

**Summary:**

The paper proposes a novel message-passing paradigm in which nodes have different states and can decide to receive, send, and ignore messages based on the states. The states are determined by the output of a Gumbel-softmax estimator trained along with the model. Some theoretical analysis is provided to show that the proposed method alleviates the over-squashing issue that regular GNNs have.

**Strengths:**

The paper is generally well-written and easy to follow. The motivation for using such a model to prevent over-squashing is explained well, both theoretically and intuitively. The proposed model effectively reduces the total number of message passing, which potentially improves efficiency. However, the actual improvement is unclear due to the non-deterministic nature of the method. The analysis of the kept edges across layers provides good insights into the underlying mechanism of the technique.

**Weaknesses:**

My primary concern lies in that this work appears very similar to Agent-based Graph Neural Networks [1]. An agent that decides to move towards neighboring nodes can be seen as listen/broadcast states in CoGNN, and an agent that decides to stay in the current node can be seen as the isolate state in CoGNN. In such a case, will CoGNN be identical to Agent-based Graph Neural Network?

The graph classification datasets being evaluated are small compared to larger datasets such as QM9 and ZINC.

Node classification evaluation is only conducted on heterophilic datasets. While cora is evaluated in the appendix, showing CoGNN generally works on homophilic graphs is also essential.

[1] Martinkus, Karolis, et al. "Agent-based graph neural networks." arXiv preprint arXiv:2206.11010 (2022).

**Questions:**

- While it makes sense that for homophilic graphs, less directed edges in the last layers help filter distant nodes' information, it does not make much sense to me why fewer directed edges in the first layers help heterophilic graphs learning. It is essentially suppressing the first few layers, but the last layers are still promoting homophilic learning. Can you explain this further?

- Can you explain the difference between CoGNN and agent-based GNN?

- How is gradient propagated to the Gumbel-softmax estimator?

---

> ### Author Response · Authors · 2023-11-16
> **Rebuttal Reviewer 9qTL (Part 1 of 2)**
>
> We thank the reviewer for their comments and for finding our paper well-written and well-explained both intuitively and theoretically. We answer the concerns raised by the reviewer  below:
>
> > Q1. While it makes sense that for homophilic graphs, less directed edges in the last layers help filter distant nodes' information, it does not make much sense to me why fewer directed edges in the first layers help heterophilic graphs learning. It is essentially suppressing the first few layers, but the last layers are still promoting homophilic learning. Can you explain this further?
>
> This is an intricate aspect of Co-GNNs that we are happy to elaborate further. On heterophilic datasets, the features of direct neighbors are not necessarily informative. Therefore, the action network follows a rather *cautious* approach in the early layers, by focusing only on the most informative nodes. This allows the informative node features to be propagated across the graph, which eventually leads to obtaining more informative features for the other nodes. As a result of these changes in the node features, the action network gradually enables information propagation between more nodes. We can see this in the visualization of the *Minesweeper* experiment (**Sec 6, response.pdf**): Even though the game necessitates communication between the neighboring nodes, the action networks initially focus on the most informative nodes and only after updating the other node features with valuable information, it starts enabling communication between these.
>
> > Q2. Can you explain the difference between CoGNN and agent-based GNN?
>
> > Related comment: My primary concern lies in that this work appears very similar to Agent-based Graph Neural Networks . An agent that decides to move towards neighboring nodes can be seen as listen/broadcast states in CoGNN, and an agent that decides to stay in the current node can be seen as the isolate state in CoGNN. In such a case, will CoGNN be identical to Agent-based Graph Neural Network?
>
> Thanks for pointing out this related work: we will include a detailed discussion about this in our paper. In a nutshell: Co-GNNs and Agent-based GNNs are generally incomparable in terms of their capabilities and properties. In more detail:
>
> **Conceptual differences:** In Agent-based GNNs, several agents 'walk' on the graph nodes and collect information from the node states. The information collected by these agents is then used to update the node representations at every layer. Conceptually, we think this work is more closely related to random walk approaches. Differently, Co-GNNs do not have any agents walking on the graph, and every node is viewed as a player.
>
> **Differences in the updates:** Agent-based GNNs update a node representation only if there is an agent on this node. If there is an agent on the node then they update the node based on *all* neighbors of the node, following a standard message-passing paradigm. Differently, Co-GNNs allow intricate ways of updating the node features based on subsets of neighbors induced by the various action configurations of neighboring nodes. In agent-based GNNs, if there is no agent on the node then the node retains its state in agent-based GNNs. This is *not identical* to our ISOLATE either, because an isolated node in Co-GNNs still updates its representation based on its previous state (node-wise update) rather than retaining its state.
>
> **Differences in the approach:** In their current form, agent-based GNNs are limited to graph-based tasks. In fact, if a node is never visited by an agent then this node representation will *not* be updated: The original paper uses a number of agents which grows with the mean of graph sizes to ensure the graph is sufficiently traversed. Moreover, the action network in Co-GNNs can be viewed as a *look-ahead network* that makes decisions after 'applying' $k$ GNN layers. It can therefore recognize, before the environment network aggregates any information, the relevant information based on the $k$-hop nodes. This is a unique feature of Co-GNNs, which allows the aggregation mechanism to be conditioned on this look-ahead capability, which is another difference from agent-based GNNs.

---

> ### Author Response · Authors · 2023-11-16
> **Rebuttal Reviewer 9qTL (Part 2 of 2)**
>
> > Comment: Node classification evaluation is only conducted on heterophilic datasets. While cora is evaluated in the appendix, showing CoGNN generally works on homophilic graphs is also essential.
>
> Following the reviewer’s (and **reviewer nWqM’s**) suggestion, we included experiments on classical datasets *Cora* and *Pubmed* (**Table 1, response.pdf**). These datasets are highly homophilic and therefore they are not the most suitable choices for evaluating the information flow, which is why we excluded them initially. Interestingly, however, Co-GNNs always lead to improvements also on these datasets compared to their environment/action network architectures, leading to competitive results overall. We also conduct additional experiments which can be found in **response.pdf**.
>
> > Comment: How is gradient propagated to the Gumbel-softmax estimator?
>
> The Gumbel-Softmax uses a continuous relaxation of the discrete sampling operation, making it differentiable. This is achieved by drawing samples from a Gumbel distribution and subsequently applying a softmax function to obtain the probabilities. Co-GNNs use a variation of the Gumbel-Softmax estimator called the Straight-through Gumbel-Softmax estimator, which performs discretized sampling but uses continuous relaxation in the backward pass. We discuss these details in Appendix E.1, but more details can be found in Section 2 of the original work [1].
>
>
> > Comment: The graph classification datasets being evaluated are small compared to larger datasets such as QM9 and ZINC.
>
> Following the reviewer’s suggestion, we included experiments on *ZINC*, a classical molecular screening dataset (**Table 2, response.pdf**). The objective of this task strongly correlates with cycle counts (rings in organic molecules), so those subgraph GNNs that explicitly inject such information will perform strongly. For fairness, we only considered GNN architectures that do not inject such subgraph counts. From **Table 2, response.pdf,** Co-GNNs outperform all architectures, including GAT [2], GatedGCN [3], and PNA [4].
>
> *[1] Jang E., Gu S., and Poole B. "Categorical reparameterization with gumbel-softmax", ICLR 2017.*
>
> *[2] Velickovic P., Cucurull G., Casanova A., Romero A., Lio P., and Bengio Y. Graph attention networks. In ICLR, 2018.*
>
> *[3] Bresson X. and Laurent T. “Residual gated graph convnets”.*
>
> *[4] Corso G., Cavalleri L., Beaini D., Lio P., and Velickovic P. ”Principal neighbourhood aggregation for graph nets”. NeurIPS 2020.*
>
>
> We believe to have addressed all the concerns of the reviewer, and hope they will increase their score. We will be happy to answer any further questions.

---

> ### Comment · Reviewer_9qTL · 2023-11-19
>
> The author clarified most of my confusion. While the answer to the model's exceptional performance on heterophilic graph still appears hand-wavy to me, I very much appreciate the minesweeping example, which helps me understand the method better. Hence, Ive updated my rating from 5 to 6, and prone to acceptance.

---

> > ### Author Response · Authors · 2023-11-20
> >
> > We thank the reviewer for going through our rebuttal and raising their score. We are happy to hear that their concerns have been largely addressed by our rebuttal. Indeed, Co-GNNs have exceptionally strong performance on heterophilic datasets. However, since the target functions to be learned are not explicitly known, it is hard to precisely pinpoint the reasons behind the decisions of the action network (i.e., we do not know the ideal/optimal actions to start with). On the other hand, Minesweeper better lends itself to interpretability, since it is based on a (structured) game on a grid with precise rules: we are glad the reviewer appreciated this visual analysis and leans towards acceptance. Please let us know if there are any open concerns/questions: we are happy to clarify.

---

### Official Review · Reviewer_nWqM · 2023-10-31

**Soundness:** 2 fair
**Presentation:** 3 good
**Contribution:** 2 fair
**Rating:** 3
**Confidence:** 5

**Summary:**

The authors of this paper propose a new message-passing scheme where each node can lister or broadcast messages, and therefore can alter the graph topology by removing or making some edges directed.

**Strengths:**

1. The proposed model is flexible in adapting the graph topology to specific tasks. By allowing nodes to choose their message-passing actions based on their state, the model can optimize information flow according to the task at hand. This task-specific adaptation is a valuable feature, as it enables better exploration of graph topologies.

2. The paper is well-written and the idea is communicated very clearly.

**Weaknesses:**

1. Novelty and Clarifications: The proposed approach may lack technical novelty as it primarily involves altering the direction of edges or discarding them altogether.  Additionally, the paper does not provide a clear explanation of how the proposed approach can be adapted for use in directed graphs, where edge direction is a critical factor, leaving a gap in its applicability.
Furthermore, the authors claim that the model can efficiently filter out irrelevant information by focusing on the shortest path connecting two nodes to maximize information flow to the target node. However, due to the node-based nature of the actions, there is a significant risk that a node, for example, u_1, may be irrelevant for one node, u_2, but important for another node, u_3. With the current framework, if node u_1 chooses to listen and not broadcast, it won't be able to transmit information to u_3, creating a potential limitation in information flow within the model.
In contrast, Graph Attention Networks (GATs) can address similar challenges by focusing on edge-based attention scores, allowing for fine-grained control over information transmission. GATs can effectively filter out irrelevant information by learning attention scores close to zero, similar to the "listen and not broadcasting" actions mentioned in the proposed framework. A thorough discussion of the similarities and differences between the proposed approach and GATs, as well as other relevant models, would enhance the paper's clarity and its position in the context of existing graph neural network research.

2. Expressivity: Since the sampling process can introduce variability, it's possible for two identical graphs to obtain different representations due to the differences in sampled action. While this non-deterministic behavior enhances expressiveness, it's essential to be aware of it when working with CO-GNNs, as it may introduce variability in model outcomes and representations, even for isomorphic graphs. A proper discussion of this limitation is currently missing.

3. Experiments: The experimental evaluation is weak. Several strong baselines are missing (see below for some examples [1,2,3] ) and some well-known graph and node classification datasets are missing as well (REDDIT-BINARY, REDDIT-MULTI, Cora, Citeseer, Texas, Wisconsin, Cornell etc). Moreover,  the proposed approach does not significantly outperform the simple GIN model in most cases.

References:

[1] Zhang, Muhan, and Pan Li. "Nested graph neural networks." Advances in Neural Information Processing Systems 34 (2021): 15734-15747.

[2] Pasa, L., Navarin, N. & Sperduti, A. Polynomial-based graph convolutional neural networks for graph classification. Mach Learn 111, 1205–1237 (2022). https://doi.org/10.1007/s10994-021-06098-0

[3] Nikolentzos, Giannis, Michail Chatzianastasis, and Michalis Vazirgiannis. "Weisfeiler and Leman go Hyperbolic: Learning Distance Preserving Node Representations." International Conference on Artificial Intelligence and Statistics. PMLR, 2023.

**Questions:**

1. Can the authors elaborate on how the proposed approach can be adapted for use in directed graphs? Edge direction is crucial in various real-world applications, and it would be valuable to understand how the model deals with it.

2. The paper claims that the model can efficiently filter out irrelevant information by focusing on the shortest path. However, it's not clear how this works when information from one node is important for some nodes and irrelevant for others (see weaknesses above). Could the authors provide more clarity on this aspect?

3. The paper introduces an innovative approach, but it's important to discuss its similarities and differences with existing models, such as Graph Attention Networks (GATs) (see weaknesses above). A thorough discussion would help readers understand the model's unique contributions. Can the authors provide insights into how their approach compares to GATs or other models in terms of filtering out irrelevant information and optimizing information flow within a graph?

4. It's mentioned that CO-GNNs introduce variability due to the non-deterministic nature of the sampling process. Could the authors discuss the potential limitations and implications of this variability, especially when working with real-world data or applications where consistency in representations is critical?

5. The experimental evaluation lacks some strong baselines and omits some well-known graph and node classification datasets. Could the authors explain the rationale for the choice of datasets and provide justification for not including certain well-known benchmarks like REDDIT-BINARY, REDDIT-MULTI, Cora, Citeseer, Texas, Wisconsin, Cornell, etc.?

---

> ### Author Response · Authors · 2023-11-16
> **Rebuttal Reviewer nWqM (Part 1 of 3)**
>
> We thank the reviewer for their comments and for finding the idea of the task and graph-specific information flow valuable and the writing of the paper very clear! We answer each of the raised questions below:
>
> > Q1: Can the authors elaborate on how the proposed approach can be adapted for use in directed graphs? Edge direction is crucial in various real-world applications, and it would be valuable to understand how the model deals with it.
>
> > Related comment: The paper does not provide a clear explanation of how the proposed approach can be adapted for use in directed graphs, where edge direction is a critical factor, leaving a gap in its applicability.
>
> While we focus on simple, undirected graphs in our paper, there is no fundamental limitation in adapting Co-GNNs to directed, and even multi-relational graphs. We think this is an important future direction, but one that requires a dedicated, separate study. One possible adaptation is by including actions that also consider the directionality: For example, for each node $u$, we can define the actions LISTEN-INC (listen to nodes that have an incoming edge to $u$) and LISTEN-OUT (listen to nodes that have an incoming edge from $u$) and extend the other actions analogously to incorporate directionality [1]. Note that similar ideas could be applied to even multi-relational graphs.  However, each of these design choices requires a thorough study which is beyond the scope of the current work. Another approach is to use action/environment networks that can adequately handle directed (or multi-relational) graphs (directional GNNs): In this case, we can directly use Co-GNNs with these action/environment architectures, and we do not anticipate any changes in the action space. We will include a discussion on these avenues of future directions. Thank you for pointing this out!
>
>
> >Q2: The paper claims that the model can efficiently filter out irrelevant information by focusing on the shortest path. However, it's not clear how this works when information from one node is important for some nodes and irrelevant for others (see weaknesses above). Could the authors provide more clarity on this aspect?
>
> >Related comment: The authors claim that the model can efficiently filter out irrelevant information by focusing on the shortest path connecting two nodes to maximize information flow to the target node. However, due to the node-based nature of the actions, there is a significant risk that a node, for example, $u\_1$, may be irrelevant for one node, $u\_2$, but important for another node, $u\_3$. With the current framework, if node $u\_1$ chooses to listen and not broadcast, it won't be able to transmit information to $u\_3$, creating a potential limitation in information flow within the model.
>
> This is a very subtle point. Technically, node-based actions can capture all topological configurations that can be obtained by edge-based actions using sufficiently many layers. For example, suppose we are interested in the edges $(u\_1, u\_2) $, $(u\_1,u\_3)$ and the node state $u\_1$ is only relevant for $u\_3$ but not for $u\_2$. This can be achieved if $u\_1$ broadcasts, $u\_2$ isolates, and $u\_3$ listens. It is slightly trickier if $u\_2$ has another neighbor whose information needs to be transmitted to $u\_2$. In this case, we can 'serialize' the process: after applying one layer as before, we can now isolate $u\_1$ and allow $u\_2$ to listen. The examples can be made more complex, but the main idea is always the same. In this sense, the node-based actions are powerful. Our framework can trivially be extended to edge-based actions, which could lead to more succinct constructions, but with the downside of having a potentially much larger actions space (and higher runtime complexity). Therefore, we focused on node-based actions in our study.

---

> ### Author Response · Authors · 2023-11-16
> **Rebuttal Reviewer nWqM (Part 2 of 3)**
>
> > Q3: The paper introduces an innovative approach, but it's important to discuss its similarities and differences with existing models, such as Graph Attention Networks (GATs) (see weaknesses above). A thorough discussion would help readers understand the model's unique contributions. Can the authors provide insights into how their approach compares to GATs or other models in terms of filtering out irrelevant information and optimizing information flow within a graph?
>
> > Related comment: In contrast, Graph Attention Networks (GATs) can address similar challenges by focusing on edge-based attention scores, allowing for fine-grained control over information transmission. GATs can effectively filter out irrelevant information by learning attention scores close to zero, similar to the "listen and not broadcasting" actions mentioned in the proposed framework.
>
> > Related comment: A thorough discussion of the similarities and differences between the proposed approach and GATs, as well as other relevant models, would enhance the paper's clarity and its position in the context of existing graph neural network research.
>
> This is a very good point. Let us highlight the fundamental differences:
> * GATs can learn appropriate attention coefficients to filter out some information from the neighbors but learning the $0$ coefficient (i.e., discard the message altogether) is typically hard in practice (soft attention), whereas the action network in Co-GNNs can easily choose the action "isolate" (as widely observed in our experiments).
> * Suppose for the sake of the argument that GATs can learn the best attention coefficients. This does not always yield fine-grained control over the information flow. The argument hinges on the fact that attention is feature-based and it is normalized via softmax. If GAT attends to a particular node in the neighborhood then it must attend to all other neighbors who have identical features in the exact same way. This causes the following effect: Fix a GAT model and apply it to a test graph that has higher degree nodes that also have more neighbors having identical features. Then, the contribution of these identical node features increases with their frequency (and this effect cannot be avoided), eventually belittling the contribution of features of all other nodes. Intuitively, this is due to softmax normalization, which produces smaller scores for less frequent node features.
> * GATs have other inherent limitations. For example, they cannot even detect node degrees. This is evident in our *RootNeighbors* experiment, where the information needs to be propagated only from degree-6 nodes, and GAT model, unable to detect node degrees, performs poorly. In other words, GATs may not be able to detect which information needs to be filtered (especially structural ones).
> * The contribution of Co-GNNs is *orthogonal* to that of GATs. In fact, GATs can be used as a base architecture in Co-GNNs. Our message-passing paradigm generally increases the performance of GATs. To show this, we experiment with CoGNN($\Sigma$,$\alpha$) in our synthetic example, which results in an 80% decrease in MAE (**Table 5, response.pdf**), from the initial GAT performance which was basically a random guess. In this case, the action network allows GAT to determine the right topology, and GAT only needs to learn to average the features.
> * The action network of Co-GNNs can be viewed as a look-ahead network that makes decisions after 'applying' $k$ GNN layers. It can therefore recognize, before the environment network aggregates any information, the relevant nodes based on the $k$-hop information. This is a unique feature of Co-GNNs, which allows the aggregation mechanism to be conditioned on this look-ahead capability, which is not present in GATs.

---

> ### Author Response · Authors · 2023-11-16
> **Rebuttal Reviewer nWqM (Part 3 of 3)**
>
> > Q4: It's mentioned that CO-GNNs introduce variability due to the non-deterministic nature of the sampling process. Could the authors discuss the potential limitations and implications of this variability, especially when working with real-world data or applications where consistency in representations is critical?
>
> > Related comment: Expressivity: Since the sampling process can introduce variability, it's possible for two identical graphs to obtain different representations due to the differences in sampled action. While this non-deterministic behavior enhances expressiveness, it's essential to be aware of it when working with CO-GNNs, as it may introduce variability in model outcomes and representations, even for isomorphic graphs. A proper discussion of this limitation is currently missing.
>
> We noted this as a limitation (on Page 6, Section 5 in our submission), and we are happy to elaborate further as per your request. Let us give more insights about this: Co-GNNs have standard deviations (over different benchmarks) which are typically smaller than most other models. This means that despite the non-deterministic nature of the algorithm, the model is generally robust and stable. We think this is a consequence of having skewed probability distributions for actions. In fact, as we decrease the temperature, the gumble-softmax vectors closely tend to a 1-hot vector, minimizing the variance.
>
> > Q5: The experimental evaluation lacks some strong baselines and omits some well-known graph and node classification datasets. Could the authors explain the rationale for the choice of datasets and provide justification for not including certain well-known benchmarks like REDDIT-BINARY, REDDIT-MULTI, Cora, Citeseer, Texas, Wisconsin, Cornell, etc.?
>
> * Platanov et al. (2023) showed that the datasets *Texas*,*Wisconsin*,*Cornell* have serious drawbacks, such as extreme class imbalance, duplicates, etc. This makes the results obtained on these datasets unreliable and we opted for using the most recent and challenging heterophilic dataset proposed by Platanov et al. (2023). Co-GNNs show a great improvement in heterophilic datasets, outperforming all competitive architectures, including variations of attention networks (e.g., graph transformers), despite only relying on SumGNNs and MeanGNNs.
> * We did conduct experiments on *REDDIT-B* and these are already present in our submission in Table 3.
> * Following your suggestion, we included experiments on *Cora* and *Pubmed* (**Table 1, response.pdf**). These datasets are highly homophilic and therefore they are not the most suitable choices for evaluating the information flow,  which is why we excluded them initially. Interestingly, however, Co-GNNs always lead to improvements also on these datasets compared to their environment/action network architectures, leading to competitive results overall. We also conduct additional experiments on *REDDIT-M* (**Table 4, response.pdf**), based on the feedback all of which is reported in **response.pdf.**
> * We view Co-GNNs as a framework rather than a single model architecture, and all the presented results can potentially be improved, by using other architectures as action/environment networks. In fact, Co-GNNs can be used with most existing architectures without any modifications. We intentionally opt for simple architectures to show that the virtue of our approach lies in the new message-passing paradigm rather than potential confounding factors arising from the power of base architectures. We kindly ask the reviewer to evaluate our empirical results in the light of this. We achieve state-of-the-art results on the heterophilic graphs using these simple architectures as action/environment networks.
>
> *[1] Rossi E., Charpentier B., Di Giovanni F., Frasca F., Günnemann S. and Bronstein M. “Edge Directionality Improves Learning on Heterophilic Graphs”, LoG 2023.*
>
> *[2] Platonov, O. and Kuznedelev, D. and Diskin, M. and Babenko, A. and Prokhorenkova, L. "A critical look at the evaluation of GNNs under heterophily: are we really making progress?", ICLR 2023.*
>
> We believe to have addressed all the concerns of the reviewer, and hope they will increase their score. We will be happy to answer any further questions.

---

> ### Author Response · Authors · 2023-11-21
>
> **Dear reviewer nWqM:** Thank you for raising your concern about the similarities and dissimilarities between GAT and CoGNNs. We believe to have addressed all of your concerns. As the rebuttal discussion is closing soon, we would highly appreciate feedback at this stage since this would give us a last chance to address any remaining issues. We look forward to hearing from you. Thank you.

---

> > ### Comment · Reviewer_nWqM · 2023-11-22
> >
> > I would like to thank the authors for their response and the new experiments. However, most of my concerns still remain.
> > Specifically, regarding Q2,  you mention that you can solve the problem by 'serializing' the process and applying many layers. However, this does not solve the case when information should be exchanged in the same layer.
> > Moreover, I understand that Co-GNN is a framework and can be combined with many GNN architectures. However, I do not see any significant improvement in most cases (for example Co-GNN(Σ,Σ) vs GIN). Therefore, I will retain my initial score.

---

> ### Author Response · Authors · 2023-11-22
>
> We thank the reviewer for following up on our rebuttal. In the review, there were 5 questions/concerns and we carefully went through all of them. We follow up on the items reviewer has raised:
> * **Empirical improvements:** We would like to highlight the substantial improvements obtained on heterophilic graphs (SOTA results). These datasets illustrate the potential of Co-GNNs very strongly.  Indeed the promise of the paper lies mostly in heterophilic graphs where information propagation is more subtle. We also see minor improvements on homophilic datasets, which is perfectly in line with the presented theory. Furthermore, we kindly ask the reviewer to take into account the two additional experiments from the rebuttal period, precisely addressing the concern:
>     * **ZINC experiment and GIN:** In this experiment, Co-GNNs improve substantially on all baselines. In fact,  Co-GNN($\Sigma,\Sigma$) reduces MAE by **31.7**% compared to GIN (**Table 2, response.pdf**).
>     * **RootNeighbors and GAT:** In this experiment, we show that using GATs as part of Co-GNNs leads to great improvements on our synthetic dataset, improving GATs random guess behaviour leading to a strong model, reducing MAE by **~80**% (**Table 5, response.pdf**).
> * **Remaining concern regarding Q2:** It is correct that a single layer parametrization is not always sufficient in Co-GNNs, but this does not pose a problem, because (1) there is no information loss overall, i.e., technically the composition of k layers will achieve exactly the same update that reviewer asked, and (2) we can build deeper Co-GNNs thanks to the refined information flow.
> * **GATs vs CO-GNNs:** The original review stated the need for a detailed discussion comparing GATs and Co-GNNs as an important concern. We believe to have addressed this concern thoroughly. We are happy to elaborate further if anything remains unclear.
>
> Finally, we are happy to add the baselines (as well as other baselines) pointed in the review, but as we stressed earlier, these baselines themselves can be used as part of the Co-GNN framework and fundamentally do not affect the message of this paper.

---

### Official Review · Reviewer_7kxJ · 2023-11-02

**Soundness:** 3 good
**Presentation:** 4 excellent
**Contribution:** 4 excellent
**Rating:** 8
**Confidence:** 3

**Summary:**

In this paper, the author proposed a more flexible and dynamic message-passing paradigm. In this paradigm, each node is viewed as a player with actions of either 'listen', 'broadcast', 'listen and broadcast' or 'isolate'. Based on this paradigm, the author proposed a new GNN, called cooperative GNN (Co-GNNs). Experimental results on regression, node classification, graph classification have showed better or competitive performance than SOTA works.

**Strengths:**

I have enjoyed reading the manuscript because it proposed a novel and solid message passing operation. The introduction is written very well, that is easy to make readers easily immersed in this work. The author has done detailed theoretical analysis and experimental analysis to prove the superiority of the proposed Co-GNN framework. I believe this work can inspire many subsequent works, making potential contributions to the GNN field.

**Weaknesses:**

My only concerns is if it is possible to shows some visualization results for this new message passing operation on the node- and graph-level task,e.g., For a right prediction, how each nodes behaves in the Co-GNN framework.  I think that will make readers more appreciate this nice work, and will encourage following works along this line.

**Questions:**

Please see the above

---

> ### Author Response · Authors · 2023-11-16
> **Rebuttal Reviewer 7kxJ**
>
> We thank the reviewer for their comments and for finding our work "novel, solid, inspiring, and well-presented". We very much agree with the raised concern regarding the visualization of the actions:
>
> > Comment: My only concerns is if it is possible to shows some visualization results for this new message passing operation on the node- and graph-level task,e.g., For a right prediction, how each nodes behaves in the Co-GNN framework. I think that will make readers more appreciate this nice work, and will encourage following works along this line.
>
> Following this feedback, we produced a new visualization using the trained Co-GNN model on *Minesweeper* (**Sec 6 in response.pdf**): *Minesweeper* (Platanov et al., 2023) is a heterophilic dataset inspired by the Minesweeper game. The graph is a regular 100x100 grid where each node (cell) is connected to eight neighboring nodes, where 20% of the nodes are randomly selected as mines. The task is to predict which nodes are mines. The node features are one-hot-encoded numbers of neighboring mines. However, for randomly selected 50% of the nodes, the features are unknown, which is indicated by a separate binary feature.
>
> **Visualization setup.** We take a trained $10$-layer Co-GNN model and present the evolution of the graph topology from layer $\ell=1$ to layer $\ell=8$ (**Sec 6 in response.pdf**). We choose a node (black), and at every layer $\ell$, we depict its neighbors up to distance $10$.  In this visualization, nodes which are mines are shown in red, and other nodes in blue. The features of non-mine nodes (indicating the number of neighboring mines) are shown explicitly whereas the nodes whose features are hidden are labeled with a question mark. For each layer $\ell$, we gray out the nodes whose information cannot reach the black node with the remaining layers available.
>
> **Visualization results.** Interestingly, in the early layers, the action network learns to isolate the right section of the black node, similar to how humans would play this game: The bulk of the nodes without neighboring mines ($0$ labeled nodes) initially do not help in determining whether the black node is a mine or not. Thus, the action network prioritizes the information flowing from the left sections of the grid where more mines are present. We find this very nice and informative: Action network initially focuses mostly on nodes that are more informative for the task. After identifying the most crucial information and propagating this through the network, it then requires this information to also be communicated with the nodes that initially were labeled with $0$. This leads to an almost fully connected grid in the later layers $\ell$=$7,8$.
>
> Related to the visualization question, we also want to draw the reviewer’s attention to our Figure 6 of *RootNeighbors*: Co-GNN models decide to 'keep or drop' the edges accurately in >99% of the cases. Figure 6 shows an *actual* sample from the test set: The trained Co-GNN model extracted the tree shown in Figure 6 (below) from a test tree shown in Figure 6 (above). This shows that Co-GNNs can capture the optimal behavior in this case.
>
> Moreover, we provide a detailed discussion on visualizing the actions in Appendix B, where we opted for presenting the ratio of the directed edges that are kept across the different layers in Figure 7. The interesting finding here is relative to homophilic and heterophilic datasets: On *Cora* (homophilic) the ratio of edges that are kept gradually decreases as we go to the deeper layers, and on *roman-empire* (heterophilic), the ratio of edges that are kept gradually increases after layer 1. This contrasting behavior suggests that the CO-GNN model can adapt to the nature of the data, by changing the flow of information.
>
> We believe to have addressed all the concerns of the reviewer, and hope they will increase their score. We will be happy to answer any further questions.

---

> ### Author Response · Authors · 2023-11-21
>
> **Dear reviewer 7kxJ:** Thank you for raising your concern about the need for a visualization of our message-passing paradigm. We believe to have addressed all of your concerns. As the rebuttal discussion is closing soon, we would highly appreciate feedback at this stage since this would give us a last chance to address any remaining issues. We look forward to hearing from you. Thank you.

---

> > ### Comment · Reviewer_7kxJ · 2023-11-22
> >
> > Thanks for providing a visual representation of the Minesweeper game. In fact, I feel that it would be better if visualizations for some tasks mentioned in the original paper could be included, just as done in this work: https://arxiv.org/pdf/2106.05667.pdf
> >
> > Although I note that other reviewers have a negative attitude towards this paper, I am still inclined to accept it and believe it has significant value. If there is a rating choice of 9 point, I will raise my evaluation to that. Unfortunately, I have to maintain my rating as it is, i.e., 8 point.

---

> ### Author Response · Authors · 2023-11-22
>
> We thank the reviewer for their continued support. We will make sure to include the additional visuals requested by the reviewer in the final version of the paper and we are currently working hard to conclude them before the rebuttal period closes.

---

### Official Review · Reviewer_8ypJ · 2023-11-02

**Soundness:** 2 fair
**Presentation:** 2 fair
**Contribution:** 2 fair
**Rating:** 5
**Confidence:** 4

**Summary:**

The paper considers GNNs that are typically based on a nearest neighbor message passing protocol, and specializes to include learning how nodes should communicate.  This is set up as an action space with 4 possibilities, and the action space is co-learned for each node along with the associated GNN, and refer to this a cooperative-GNNs (Co-GNNs) approach.  The Gumbel-softmax is used to train the action network as part of the co-learning.  The idea could in principle be applied to many GNNs to co-learn.  Some experiments show that the Co-GNN idea may have merit in some scenarios, although the results are mixed.

**Strengths:**

The basic idea is interesting, and potentially provides a useful GNN tool for cases where graph isomorphism needs to be characterized, e.g., graph classifiers.

The method has the potential to reduce inference computational complexity.

The method may reveal long range dependencies that aren’t easily described with conventional GNNs.

The co-learning appears to be a straightforward addition to existing GNN architectures, so this opens the idea of how to harness this for different scenarios, some of which are explored in the paper.

**Weaknesses:**

The paper seems to be preliminary; interesting but with too many loose ends.  The experiments are interesting, but not quite fully understood yet.

The actions lead to a time-varying topology, but the results are not connected to the many graph theoretical works on random graphs, or shortest-path routing protocols in communications networks. The Appendix B results are interesting, and clearly suggest an analysis based on information flows within the graph.

Theorem 5.2 is interesting, but ultimately seems only to say that there will (or can) be a path learned between arbitrary nodes for them to transfer information.

The method seems likely to be brittle over topology or some erroneous training data, and it isn’t clear what kind of generality is possible.

The synthetic example in section 6.1 obviously favors the proposed method and the problem is very artificial.  A useful solution would be model based.

Ablation studies are needed to better understand the benefits and issues.

*Revised Review*

The authors have addressed some of my questions and issues, although there remains a considerable focus on discussion with mixed results, and the synthetic example is just that.

**Questions:**

Section 5.3: Is there a claim that the method will always learn a shortest-path route?  For example, it isn’t clear what this means of information is acted upon along the way.

As the paper notes in Section 5.4, the actions lead to time-varying graph topology, at least in the sense of turning edges on and off in an overall fixed topology?

In the simulations there are claims about “using relatively simple architectures”, but what are the comparisons made to?

Figure 7(b).  Does the graph have specific bottlenecks or other features that restrict the flow at early layers?

---

> ### Author Response · Authors · 2023-11-16
> **Rebuttal Reviewer 8ypJ (Part 1 of 3)**
>
> We thank the reviewer for finding our co-learning setup interesting, acknowledging the promise of the framework in long-range tasks, and for pointing out its potential for empowering existing graph neural networks. We answer each of their questions/concerns below.
>
> > Q1: Section 5.3: Is there a claim that the method will always learn a shortest-path route? For example, it isn’t clear what this means of information is acted upon along the way.
>
> > Related comment: Theorem 5.2 is interesting, but ultimately seems only to say that there will (or can) be a path learned between arbitrary nodes for them to transfer information.
>
> Theorem 5.2 is in the same spirit as Lemma 1 of Topping et al. (2022), where over-squashing is characterized as the insensitivity of an $r$-layer GNN output at node $u$ to the input features of a distant node $v$. Topping et al. (2022) quantify sensitivity through a bound on the Jacobian: The higher the Jacobian $\|\partial \boldsymbol{h}\_v^{(r)}/\partial \boldsymbol{x}\_u\| \leq C^{r}(\hat{\boldsymbol{A}}^{r})\_{v u}$ the more sensitive is a node $u$ to the feature of a node $v$ (and can alleviate loss of information). Theorem 5.2 states that Co-GNNs can transmit signals from distant nodes, and, as an immediate corollary to this result, we can conclude that the Jacobian bound can be maximized. We found it more insightful to present a construction showing how such information can be propagated rather than giving a Jacobian bound, which is not necessarily informative for the reader.
>
> Theorem 5.2 does *not* imply that the information will necessarily be traveling along shortest paths. The construction for the proof is indeed by enabling information flow through a shortest path between the nodes, but this does not in any way preclude other 'routes', e.g., walks, or other substructures.  This is precisely where the strength of Co-GNNs lies: They learn to propagate information in a task-, graph-, layer-, and feature-specific manner. Given the action network, they can – but do not need to – learn to route information through paths if this is the optimal topological choice. This leads to an obvious question: How do we determine whether Co-GNNs are able to learn the optimal information propagation for a task? The optimal information propagation can be identified only if we know the target function to be learned. This is precisely the reason behind experimenting with a synthetic task of *RootNeighbors*.
>
> > Related comment: The synthetic example in section 6.1 obviously favors the proposed method and the problem is very artificial. A useful solution would be model based.
>
> As mentioned above, the objective of the synthetic experiment is to show whether Co-GNN models learn the desired information propagation mechanism. In this case, we know the target function, and we can determine which edges must be 'kept or dropped'. We verify that Co-GNN models decide to 'keep or drop' the edges accurately in >99% of the cases. Specifically, Co-GNNs had to learn to focus only on degree-6 neighbors by essentially viewing the remaining neighbors as noise for the learning task. This has been realized empirically: The trained Co-GNN model extracted the tree shown in Figure 6 (below) from a test instance shown in Figure 6 (above). We find this more insightful than the MAEs alone. This is a very simple task, where all GNNs appear to fail, and the problem of aggregating over nodes with a certain degree is implicit in real-world tasks (e.g., high-degree nodes in social networks may carry more information). We are happy to follow up if any of these points remain unanswered or unclear.

---

> ### Author Response · Authors · 2023-11-16
> **Rebuttal Reviewer 8ypJ (Part 2 of 3)**
>
> > Q2: As the paper notes in Section 5.4, the actions lead to time-varying graph topology, at least in the sense of turning edges on and off in an overall fixed topology?
>
> Indeed, the actions lead to a time-varying graph, which makes the whole message-passing approach dynamic across layers. The message-passing protocol is completely learned rather than being fixed, which is at the heart of our study.
>
> > Related comment: The method seems likely to be brittle over topology or some erroneous training data, and it isn’t clear what kind of generality is possible.
>
> We respectfully disagree. Both the action and the environment networks are GNNs that learn functions over graphs with node features. Therefore, each layer of the action network has access to the features of neighboring nodes (in deciding which actions to choose) and the actions are learned in an essentially invariant way. This makes it possible for Co-GNNs to generalize to graphs with different topologies and features. If there are 'erroneous' features in the training, then Co-GNNs can rely more on the structural information, and vice versa. Moreover, Co-GNNs can better filter out noise compared to GNNs, thanks to their learnable message-passing mechanism.
>
> This is corroborated by our synthetic experiment: Co-GNNs solve *RootNeighbors*, despite the presence of topological variations in the data (root nodes have different degrees in different trees); and the fact that most node features can be seen as noise because they are detrimental to the task as elaborated above (in response to Q1). Our empirical results on real-world heterophilic graphs (which are from different domains) present further evidence of the generalization capabilities of Co-GNNs.
>
> > Q3: In the simulations there are claims about “using relatively simple architectures”, but what are the comparisons made to?
>
> By relatively simple architectures, we refer to the use of simple architectures such as MeanGNNs and SumGNNs as action/environment networks: These models do not rely on sophisticated aggregation methods, subgraph counts, or any higher-order message passing. As the reviewer hints, we view Co-GNNs as a framework rather than a single model architecture, and all the presented results can potentially be improved, by using other, more sophisticated architectures as action/environment networks. In fact, Co-GNNs can be used with most existing architectures without any modifications. We intentionally opt for simple architectures to show that the virtue of our approach lies in the new message-passing paradigm rather than potential confounding factors arising from the power of base architectures. We kindly ask the reviewer to evaluate our empirical results in light of the fact that we are able to achieve state-of-the-art results on the heterophilic graphs using these simple architectures as action/environment networks.
>
> To assess the virtue of Co-GNNs, the reviewer points out the necessity for ablation studies. To address this, we ablated the action network in Co-GNNs. Specifically, we ran additional experiments as part of the rebuttal using MeanGNNs and SumGNNs on the heterophilic graphs (**Table 3 in response.pdf**): The empirical results suggest a clear trend leading to improvements as a result of using the action network. Similar trends are observed in other datasets and relative to other base action/environment networks (see, e.g., **Table 2 in response.pdf**). These empirical findings suggest that Co-GNNs are robust across different datasets and consistently improve the results relative to their base architectures.

---

> ### Author Response · Authors · 2023-11-16
> **Rebuttal Reviewer 8ypJ (Part 3 of 3)**
>
> > Q4: Figure 7(b). Does the graph have specific bottlenecks or other features that restrict the flow at early layers?
>
> In this context, we refer to a common distinction made between homophilic and heterophilic data. The aggregation mechanism of classical GNNs aligns with the homophily assumption: Nodes are more likely to be classified in the same way as their neighboring nodes. Classical GNNs tend to perform poorly on heterophilic data, where this assumption does not hold. The *roman-empire* dataset is a heterophilic dataset and classical aggregation can be detrimental. It was composed by [1] and has the following characteristics:
>
> *“This dataset is based on the Roman Empire article from English Wikipedia, which was selected since it is one of the longest articles on Wikipedia. Each node in the graph corresponds to one (non-unique) word in the text. Thus, the number of nodes in the graph is equal to the article’s length. Two words are connected with an edge if at least one of the following two conditions holds: either these words follow each other in the text, or these words are connected in the dependency tree of the sentence (one word depends on the other). Thus, the graph is a chain graph with additional shortcut edges corresponding to syntactic dependencies between words. The class of a node is its syntactic role (we select the 17 most frequent roles as unique classes and group all the other roles into the 18th class). For node features, we use FastText word embeddings (Grave et al., 2018).”*
>
> This means that the features of neighboring words are not necessarily syntactically informative. Therefore, the action network follows a cautious approach in the early layers (analogously to *Minesweeper*), by focusing only on *dependency* edges, which leads to more informative features for the other nodes. As a result of these changes in the node features, the action network gradually enables information propagation between neighboring words, as we increase the number of layers.  Fig 7(b) shows that Co-GNNs behave very differently than classical GNNs on *roman-empire* in terms of information propagation. On the other hand, Fig 7(a) shows the opposite trend on *Cora* which is a homophilic dataset. These contrasting trends suggest that Co-GNNs can adapt their information propagation based on the nature of the data. We will highlight this better in the main text, as we agree this analysis deserves more emphasis.
>
> *[1] Platonov, O. and Kuznedelev, D. and Diskin, M. and Babenko, A. and Prokhorenkova, L. "A critical look at the evaluation of GNNs under heterophily: are we really making progress?", ICLR 2023.*
>
> We believe to have addressed all the concerns of the reviewer, and hope they will increase their score. We will be happy to answer any further questions.

---

> ### Author Response · Authors · 2023-11-21
>
> **Dear reviewer 8ypJ:** Thank you for raising your concern about the mechanisms of CoGNNs. We believe to have addressed all of your concerns. As the rebuttal discussion is closing soon, we would highly appreciate feedback at this stage since this would give us a last chance to address any remaining issues. We look forward to hearing from you. Thank you.

---

### Author Response · Authors · 2023-11-16

We thank the reviewers for their comments. We responded to each concern in detail in our individual responses. In addition, we include a **response.pdf** to this post containing the results of all additional experiments conducted.
We provide a summary of our rebuttal based on the feedback from the reviewers:
* **New experiments on homophilic graphs:** We added experimental results on *Cora* and *Pubmed*,  which are reported in **Table 1**. (**Reviewer nWqM, Reviewer 9qTL**)
* **New experiments on graph classification:** We present additional graph classification experiment results on *ZINC* and *REDDIT-M*, reported in **Tables 2 and 4**, respectively. (**Reviewer 9qTL, Reviewer nWqM**)
* **Ablation study on heterophilic graphs:** We provided an ablation study on the heterophilic dataset to highlight the Co-GNN performance gain compared to its base architecture, reported in **Table 3**. (**Reviewer 8ypJ**)
* **GAT experiments on *RootNeighbors*:** To highlight the difference between GATs and Co-GNNs, we conducted further experiments on the synthetic dataset *RootNeighbors*, which are reported in **Table 5**. (**Reviewer nWqM**)
* **Visualization of the actions on *Minesweeper*:** We have provided a visualization of the action network on the *Minesweeper* dataset, reported in **Section 6**. (**Reviewer 8ypJ, Reviewer 7kxJ**)

*For reviewers' convenience, we included the additional findings into a separate file (**response.pdf**) but we will carefully integrate these findings into the paper after receiving further feedback from the reviewers in the coming days.*

We hope that our answers address your concerns along with the new experiments. We are looking forward to a fruitful discussion period.

**response.pdf**: https://openreview.net/attachment?id=T0FuEDnODP&name=supplementary_material

---

### Meta-Review · Area_Chair_buJU · 2024-01-07

**Metareview:**

The reviewers were split on the fate of this paper. They recognize an interesting new architecture, but were also unsure whether the experiments and baselines motivate the need for this new architecture.

**Justification For Why Not Higher Score:**

strong negative opinion

**Justification For Why Not Lower Score:**

good author response

---

### Decision · Program_Chairs · 2024-01-16

Reject